# Pandemic-scale phylogenomics reveals the SARS-CoV-2 recombination landscape

Yatish Turakhia[1,2,3,8]✉, Bryan Thornlow[1,2,8], Angie Hinrichs[2], Jakob McBroome[1,2], Nicolas Ayala[1,2], Cheng Ye[3], Kyle Smith[4], Nicola De Maio[5], David Haussler[1,2,6], Robert Lanfear[7] & Russell Corbett-Detig[1,2]✉

Accurate and timely detection of recombinant lineages is crucial for interpreting genetic variation, reconstructing epidemic spread, identifying selection and variants of interest, and accurately performing phylogenetic analyses[1-4]. During the SARS-CoV-2 pandemic, genomic data generation has exceeded the capacities of existing analysis platforms, thereby crippling real-time analysis of viral evolution[5]. Here, we use a new phylogenomic method to search a nearly comprehensive SARS-CoV-2 phylogeny for recombinant lineages. In a 1.6 million sample tree from May 2021, we identify 589 recombination events, which indicate that around 2.7% of sequenced SARS-CoV-2 genomes have detectable recombinant ancestry. Recombination breakpoints are inferred to occur disproportionately in the 3' portion of the genome that contains the spike protein. Our results highlight the need for timely analyses of recombination for pinpointing the emergence of recombinant lineages with the potential to increase transmissibility or virulence of the virus. We anticipate that this approach will empower comprehensive real-time tracking of viral recombination during the SARS-CoV-2 pandemic and beyond.

Recombination is a primary contributor of new genetic variation in many prevalent pathogens, including betacoronaviruses[6], the clade that includes SARS-CoV-2. By mixing genetic material from diverse genomes, recombination can produce new combinations of mutations that have potentially important phenotypic effects[7]. For example, recombination is thought to have played an important role in the recent evolutionary histories of Middle East respiratory syndrome[8] and severe acute respiratory syndrome coronavirus (SARS-CoV)[9-12]. Recombination might also have the potential to generate viruses with zoonotic potential in the future[13]. Therefore, accurate and timely characterization of recombination is foundational for understanding the evolutionary biology and infectious potential of established and emerging pathogens in human, agricultural and natural populations.

Now that substantial genetic diversity is present across SARS-CoV-2 populations[14] and co-infection with different SARS-CoV-2 variants has been known to sometimes occur[15], recombination is expected to be an important source of new genetic variation during the pandemic. Whether or not there is a detectable signal for recombination events in the SARS-CoV-2 genomes has been fiercely debated since the early days of the pandemic[13]. Nonetheless, several apparently genuine recombinant lineages have been identified using ad hoc approaches[16] and semi-automated methods that cope with vast SARS-CoV-2 datasets by reducing the search space for possible pairs of recombinant ancestors[16,17]. Because of the importance of timely and accurate surveillance of viral genetic variation during the continuing SARS-CoV-2 pandemic, new approaches for detecting and characterizing recombinant

haplotypes are needed to evaluate new variant genome sequences as quickly as they become available. Such rapid turnaround is essential for driving an informed and coordinated public health response to new SARS-CoV-2 variants.

We developed a new method for detecting recombination in pandemic-scale phylogenies, Recombination Inference using Phylogenetic PLacEmentS (RIPPLES, Fig. 1). Because recombination violates the central assumption of many phylogenetic methods, that is, that a single evolutionary history is shared across the genome, recombinant lineages arising from diverse genomes will often be found on 'long branches', which result from accommodating the divergent evolutionary histories of the two parental haplotypes (Fig. 1). Note that as long as recombination is relatively uncommon, phylogenetic inference is expected to remain accurate even when branch lengths are artifactually expanded[18]. RIPPLES exploits that signal by first identifying long branches on a comprehensive SARS-CoV-2 mutation-annotated tree[19,20]. RIPPLES then exhaustively breaks the potential recombinant sequence into distinct segments and replaces each onto a global phylogeny using maximum parsimony. RIPPLES reports the two parental nodes—hereafter termed donor and acceptor—that result in the highest parsimony score improvement relative to the original placement on the global phylogeny (Supplementary Text 1). Our approach therefore leverages phylogenetic signals for each parental lineage and the spatial correlation of markers along the genome. We establish significance using a null model conditioned on the inferred site-specific rates of de novo mutation (Supplementary Texts 2 and 3).

[1]Department of Biomolecular Engineering, University of California, Santa Cruz, Santa Cruz, CA, USA. [2]Genomics Institute, University of California, Santa Cruz, Santa Cruz, CA, USA. [3]Department of Electrical and Computer Engineering, University of California, San Diego, San Diego, CA, USA. [4]Department of Biological Sciences, University of California, San Diego, San Diego, CA, USA. [5]European Molecular Biology Laboratory, European Bioinformatics Institute (EMBL-EBI), Wellcome Genome Campus, Cambridge, UK. [6]Howard Hughes Medical Institute, University of California, Santa Cruz, Santa Cruz, CA, USA. [7]Department of Ecology and Evolution, Research School of Biology, Australian National University, Canberra, Australian Capital Territory, Australia. [8]These authors contributed equally: Yatish Turakhia, Bryan Thornlow. ✉e-mail: yturakhia@ucsd.edu; rucorbet@ucsc.edu

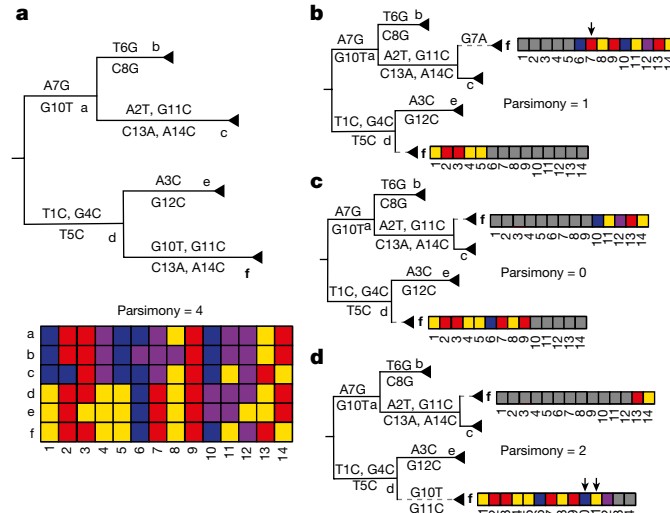

**Fig. 1 | RIPPLES exhaustively searches for optimal parsimony improvements using partial interval placements. a,** A phylogeny with six internal nodes (labelled a–f), in which node f (in bold) is the one being investigated as a putative recombinant. The initial parsimony score of node f is 4, according to the multiple sequence alignment below the phylogeny, which shows the variation among samples and internal nodes. Note that internal nodes may not have corresponding sequences in reality but test for recombination using reconstructed ancestral genomes. **b–d,** Three partial placements of the two intervals (grey cells indicate sites outside the interval) resulting from the breakpoints after site 5 (panel **b**), 9 (panel **c**) and 12 (panel **d**) respectively, along with their resulting parsimony scores. The dashed lines indicate the new branches resulting from the partial placements of f. Arrows mark sites that increase the sum parsimony of the two partial placements of f. The optimal partial placement and breakpoint prediction for node f is in the centre (**c**), with one breakpoint after site 9 and with partial placements both as a sibling of node c and as a descendant of node d.

Substantial testing via simulation indicates that RIPPLES is efficient, sensitive and can confidently identify recombinant lineages (Supplementary Texts 4–6). As expected[21], when recombination occurs towards the edges of the genome or between genetically similar sequences, it is harder to detect using RIPPLES (Extended Data Figs. 1 and 2). Nonetheless, RIPPLES detects simulated recombinants with 75.8% sensitivity. Among the simulated samples detected as recombinants, RIPPLES accurately identifies 90% of simulated breakpoints (Extended Data Table 1 and Supplementary Text 6). Furthermore, RIPPLES is able to detect all highly confident recombinants identified in a previous analysis[16] (Supplementary Text 6). Recombination analysis using RIPPLES on a global phylogeny of about 1.6 million SARS-CoV-2 genomes shows that a fraction of the sequenced SARS-CoV-2 genomes belongs to detectable recombinant lineages. To mitigate the impacts of sequencing and assembly errors, we exclude all nodes with only a single descendant, we applied conservative filters to remove potentially spurious samples from the recombinant sets flagged by RIPPLES, and we manually confirmed mutations in a subset of putative recombinant samples using raw sequence read data (Supplementary Texts 7 and 8, Extended Data Table 2 and Extended Data Fig. 3). After this, we retained 589 unique recombination events, which have a combined total of 43,104 descendant samples (Supplementary Table 1). This means that around 2.7% of total sampled SARS-CoV-2 genomes are inferred to belong to detectable recombinant lineages. Post hoc statistical analysis yields an empirical false discovery rate estimate of 11% for our statistical thresholds (Supplementary Text 9 and Extended Data Table 3). Additionally, excess similarity of geographic location and date metadata among the descendants of donor and acceptor nodes supports the notion that many ancestors of recombinant genomes co-circulated within human populations (Supplementary Texts 10 and 11 and Extended

Data Figs. 4 and 5). Because recombination events that occur between genetically similar viral lineages are challenging to detect (Extended Data Fig. 2), ours is expected to be a potentially large underestimate of the overall frequency of recombination. As a result, the RIPPLES estimate is probably conservative with respect to the global frequency of recombination in the SARS-CoV-2 population.

RIPPLES uncovered a strikingly non-uniform distribution of recombination breakpoint positions across the SARS-CoV-2 genome, consistent with previous analyses in betacoronaviruses[11,22]. In particular, among putative recombination events there is an excess of recombination breakpoints towards the 3' end of the SARS-CoV-2 genome relative to expectations on the basis of random breakpoint positions ($P < 1 \times 10^{-7}$; permutation test; Supplementary Text 12). Notably, no such bias is apparent when we simulate recombination breakpoints following a uniform distribution (Supplementary Text 13 and Extended Data Fig. 1). Change-point analysis identifies an increase in the frequency of recombination breakpoints immediately 5' of the spike protein region (20,875 base pairs; Supplementary Text 14), and this pattern is consistent when restricting ourselves to putative nodes with the largest numbers of descendants and among diverse data sources, further suggesting that it is not artefactual (Supplementary Text 15 and Extended Data Table 4). The rate of putative recombination breakpoints is about three times higher towards the 3' of the change point than the 5' interval (Fig. 2), which is similar to the relative recombination rates in the genomes of other human coronaviruses[11].

Several lines of evidence suggest that the skewed distribution of recombination breakpoint positions is not a consequence of positive selection at the level of between-host transmission dynamics. First, many of these recombinant clades have existed for a relatively short period of time, and might already be extinct. The mean timespan between the earliest and latest dates of observed descendants of detected recombinant nodes is just 37 days. Second, of the subset of recombination events that we inferred to occur between variants of concern (VOC; lineages B.1.1.7, B.1.351, B.1.617.2 and P.1 (ref. [23])) and other lineages, VOCs contribute slightly fewer spike protein mutations than non-VOC lineages on average (60 out of 125 VOC/non-VOC recombinants, $P = 0.48$, sign test). Third, recombinant clade size does not greatly differ from the remaining clade sizes, which would be expected if recombinant lineages experienced strong selection ($P = 0.8470$, permutation test). Therefore, although natural selection on between-host transmission dynamics of recombinant lineages could also impact the observed distribution of recombinant breakpoint positions[11], our data indicates that other biases shape the distribution of recombination events across the SARS-CoV-2 genome. These could include a neutral mechanistic bias affecting the distribution of recombination breakpoints.

Although not yet widespread among circulating SARS-CoV-2 genomes, recombination has measurably contributed to the genetic diversity in SARS-CoV-2 lineages. The ratio of variable positions contributed by recombination ($R$) versus those resulting from de novo mutation ($M$), $R/M$, is commonly used to summarize the relative impacts of these two sources of variation[22]. Using our dataset of putative recombination events, we estimate that $R/M = 0.00264$ in SARS-CoV-2 (Supplementary Text 16). This is low for a coronavirus population (for example, for Middle East respiratory syndrome, $R/M$ is estimated to be 0.25–0.31 (ref. [22])), which presumably reflects the extremely low genetic diversity among possible recombinant ancestors during the earliest phases of the pandemic and the conservative nature of our approach. As SARS-CoV-2 populations accumulate genetic diversity and co-infect hosts with other species of viruses, recombination will play an increasingly large role in generating functional genetic diversity and this ratio could increase[24]. RIPPLES is therefore poised to play a primary role in detecting new recombinant lineages and quantifying their impacts on viral genomic diversity as the pandemic progresses.

Our extensively optimized implementation of RIPPLES allows it to search the entire phylogenetic tree and detect recombination both within and between SARS-CoV-2 lineages without a priori defining a

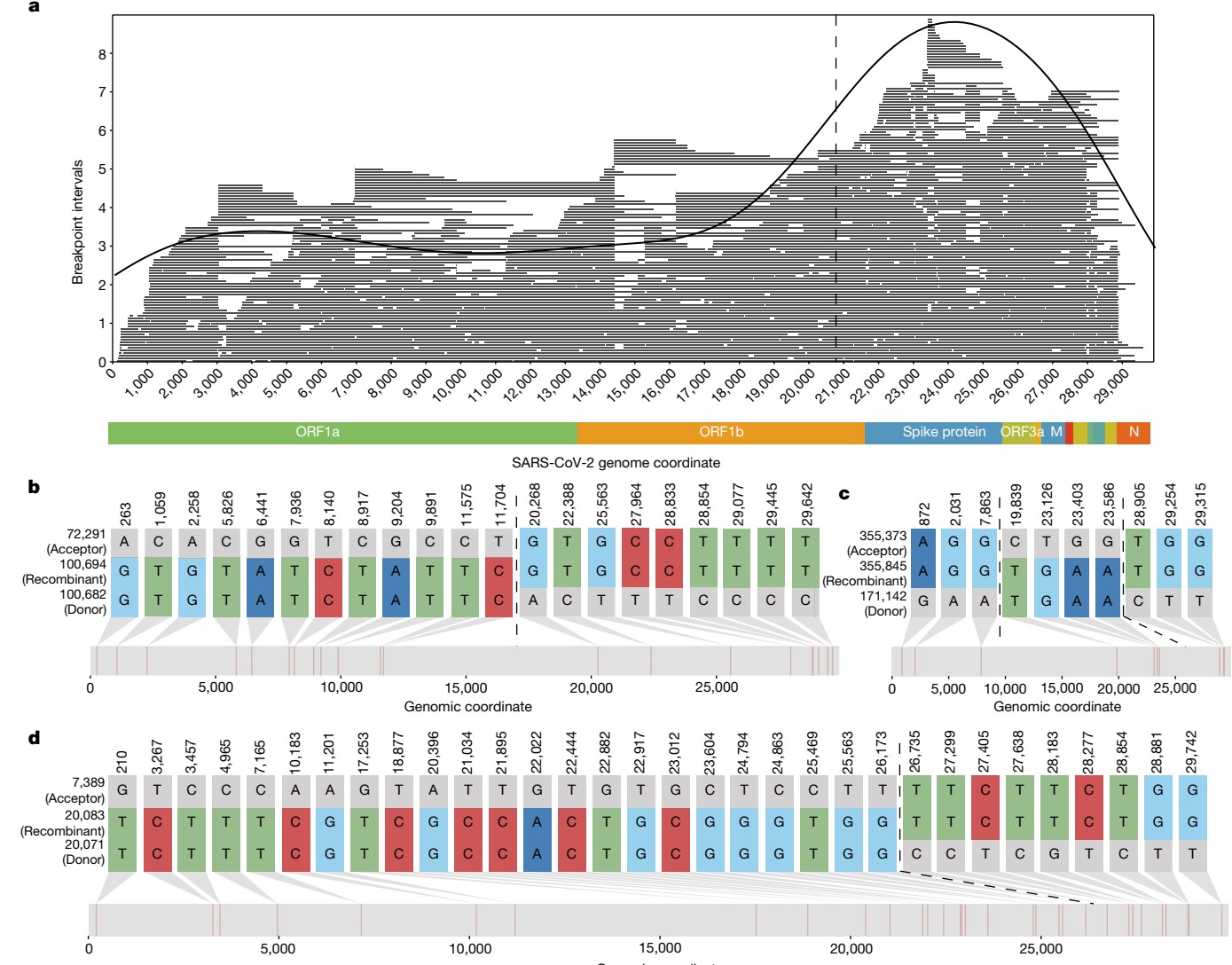

**Fig. 2 | RIPPLES detects an excess of recombination in the spike protein region. a**, The distribution of midpoints of each breakpoint's prediction interval are shown as a density plot, with the underlying recombination prediction intervals plotted as individual lines in grey. We used the midpoint of the breakpoint prediction interval because recombination events can only be localized to prediction intervals, which are the regions between two recombination-informative SNPs. A dashed vertical line at position 20,875 delimits recombination rate regions identified by change-point analysis (Supplementary Text 15). The apparent lack of recombination towards the chromosome edges probably reflects a detection bias, which we describe above (Extended Data Fig. 2). **b–d**, Recombination-informative sites (that is, positions where the recombinant node matches either but not both parent nodes) for three example recombinant trios detected by RIPPLES. The numbers to the left of each sequence correspond to the node identifiers from our MAT. **b** and **d** are examples of a recombinant with a single breakpoint (shown with dotted lines), **c** is an example of a recombinant with two breakpoints. **b–d** were generated using the SNIPIT package (https://github.com/aineniamh/snipit).

set of lineages or clade-defining mutations. This is a key advantage of our approach relative to other methods that cope with the scale of SARS-CoV-2 datasets by reducing the search space for possible recombination events (for example, refs. [16,17,25]). RIPPLES discovers 223 recombination events within branches of the same Pango lineages. Our results also include 366 interlineage recombination events (Supplementary Table 1). Additionally, we find evidence that recombination has influenced the Pangolin SARS-CoV-2 nomenclature system[23]. Specifically, we discover that the root of the B.1.355 lineage might have resulted from a recombination event between nodes belonging to the B.1.595 and B.1.371 lineages (Fig. 3 and Supplementary Table 1). These diverse recombination events highlight the versatility and strengths of the approach taken in RIPPLES.

The detection of increased recombination rates in the 3' portion of the SARS-CoV-2 genome, which contains the spike protein, highlights the utility of continuing surveillance. The spike protein is a primary location of functional novelty for viral lineages as they adapt to transmission within and among human hosts. Our discovery of both the excess of recombination events specifically around the spike protein and the relatively high levels of recombinants in circulation at present underline the importance of monitoring the evolution of new viral lineages that arise through mutation or recombination through real-time analyses of viral genomes. Our work also emphasizes the impact that explicitly considering phylogenetic networks will have for accurate interpretation of SARS-CoV-2 sequences[11].

Beyond SARS-CoV-2, recombination is a major evolutionary force driving viral and microbial adaptation. It can drive the spread of antibiotic resistance[7], drug resistance[1], and immunity and vaccine escape[2]. Identification of recombination is an essential component of pathogen evolutionary analyses pipelines as recombination can affect the quality of phylogenetic, transmission and phylodynamic inference[3]. For these reasons, computational tools to detect microbial recombination have become very popular and important in recent years[4]. The SARS-CoV-2

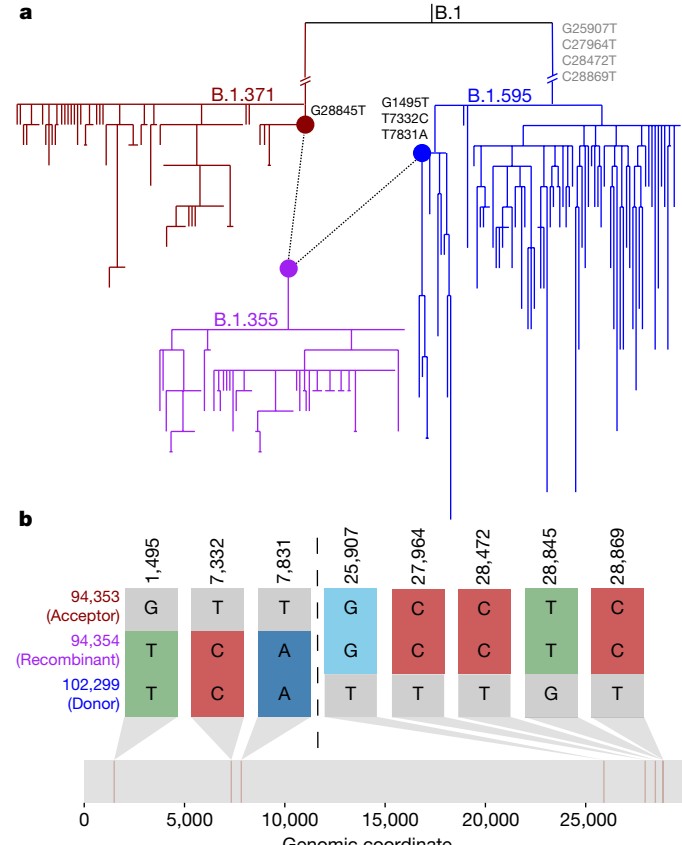

**Fig. 3 | RIPPLES uncovered evidence that the B.1.355 lineage might have resulted from a recombination event between lineages of B.1.595 and B.1.371. a**, Sub-phylogeny consisting of all 78 B.1.355 samples (purple) and the most closely related 78 samples to nodes 94,353 and 102,299 from lineages B.1.371 and B.1.595, respectively, using the 'k nearest samples' function in matUtils[20]. Nodes 94353 (red) and 102299 (blue) are connected by dotted lines to node 94,354 (purple), the root of lineage B.1.355. Recombination-informative mutations are marked where they occur in the phylogeny, with those occurring in a parent but not shared by the recombinant sequence shown in grey. **b**, Recombination-informative sites (that is, sites where the recombinant node matches either but not both parent nodes) are shown following the same format as Fig. 2b–d. **b** was generated using the SNIPIT package (https://github.com/aineniamh/snipit).

pandemic has driven an unprecedented surge of pathogen genome sequencing and data sharing, which has in turn highlighted some of the limitations of current software in investigating large genomic datasets[5]. RIPPLES was built for pandemic-scale datasets and is sufficiently optimized to exhaustively search for recombination in one of the largest phylogenies ever inferred in 40 min (Supplementary Text 17). We expect RIPPLES to perform best on densely sampled genomic datasets, which will probably become the norm for many globally distributed pathogens, but we caution that it has not yet been validated on other species. To facilitate real-time analysis of recombination among tens of thousands of new SARS-CoV-2 sequences being generated by diverse research groups worldwide each day[26–28], RIPPLES provides an option to evaluate evidence for recombination ancestry in any user-supplied samples within minutes (Supplementary Text 17). RIPPLES therefore opens the door for rapid analysis of recombination in heavily sampled and rapidly evolving pathogen populations, and provides a tool for real-time investigation of recombinants during a pandemic.

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

## Methods

RIPPLES uses the space-efficient data structure of mutation-annotated trees (MATs)[20], in which the branches of the phylogenetic tree are annotated with mutations that have been inferred to have occurred on them, to identify recombination events. Figure 1 illustrates the underlying algorithm. RIPPLES identifies putative recombinant nodes containing at least the number of mutations specified by the user and infers the set of mutations that have occurred on its corresponding sequence by accounting for all mutations annotated on the branches on its path from the root. RIPPLES then adds one or two breakpoints on mutation sites and assesses parsimony score improvement using partial placements compared to the starting parsimony. For more details, see Supplementary Text 1. To determine whether putative recombinants were significant, we developed a null model by selecting nodes at random and adding $k$ additional mutations drawn from the actual mutation spectra in our global tree. We then placed these samples on the tree and used RIPPLES to determine their parsimony score improvements (Supplementary Text 2). For each putative recombinant in our global tree, we compared its parsimony score improvement to the distribution of null parsimony score improvements for the same initial parsimony score (Supplementary Text 3). We developed our starting tree by first taking the 28 May 2021 public tree[19,20], masking all problematic sites[29], and pruning samples with fewer than 28,000 non-N nucleotides and those with two or more non-[ACGTN-] nucleotides (Supplementary Text 5). After this, we optimized this tree by running matOptimize (Supplementary Text 4) twice, with a subtree pruning and regrafting (SPR) radius of first 10 and then 40 in subsequent rounds and with the masked Variant Call Format (VCF) file as an input. Instructions for using RIPPLES are available at https://usher-wiki.readthedocs.io/en/latest/tutorials.html. We ran RIPPLES on the n2d-highcpu-224 Google Cloud Platform instance containing 224 virtual central processing units (vCPUs) (Supplementary Text 18).

To test the sensitivity of RIPPLES, we simulated recombinant samples by choosing two random internal nodes from our phylogeny with at least ten descendants and choosing breakpoints at random across the genome. We generated 1,000 simulations each for one and two breakpoint recombinants with no, one, two and three additional mutations added to the sequence after the recombination event, using scripts available at https://github.com/bpt26/recombination/. These combinations yielded 2,000 total simulated recombinant lineages. We then measured the ability of RIPPLES to detect breakpoints as a function of the position of the breakpoint and the minimum genetic distance from the recombinant node to either parent (Supplementary Text 6; genetic distance is estimated on the basis of the number of mutations inferred to separate the focal samples, lineages or nodes). We also evaluated the sensitivity of RIPPLES by ensuring that it detected each of the high-confidence recombinant SARS-CoV-2 clusters of Jackson et al.[16].

We applied several post hoc filters to remove putative recombinant nodes that may be false positives resulting from several possible sources of error. For each internal node from each trio (putative recombinant, donor and acceptor nodes) that comprised a recombinant event, we downloaded the consensus genome sequence for the nearest descendants of each node from COG-UK, GenBank, GISAID and the China National Center for Bioinformatics. We then aligned the sequences of all descendants for each trio using MAFFT[30], focusing specifically on recombination-informative sites, that is, where the allele of the recombinant node matched one parent node but not the other. If recombination-informative mutations were near to indels or missing bases, or if the entire basis for recombination was a single cluster of mutations in a 20-nucleotide span (Supplementary Text 7). We also confirmed sequence quality by manually examining raw reads for ten samples in which we could confidently link the raw sequence read data to a given consensus genome (Supplementary Text 8). To estimate the false discovery rate (FDR) associated with our specific approach and statistical threshold selected, we computed a post hoc empirical FDR. We obtained the number of internal nodes that we tested and that were associated with a given parsimony score. Then, for each initial parsimony score and parsimony score improvement, we obtained the expected number of internal nodes that would show that parsimony score improvement under the null model. Our FDR (Extended Data Table 3) is the ratio of expected nodes for a given initial and final parsimony score to the number of detected recombinant nodes with the same initial and final parsimony score (Supplementary Text 9).

We also performed post hoc analysis using sample metadata to determine whether the ancestors of the recombinant nodes had higher spatial or temporal overlap than expected by chance. We computed geographic overlap as the joint probability of choosing a sample from the same country from the descendants of the donor and the acceptor nodes. For temporal overlap, we recorded intervals from the earliest to the most recent sample descended from the donor and acceptor, respectively, and calculated the minimum number of days separating the two intervals (with 0 for overlapping intervals). We generated a null distribution for both categories by selecting, for each detected trio, two random internal nodes from the tree with a number of descendants equal to the real donor and acceptor respectively. We then calculated geographic and temporal overlap in the same way for this random set (Extended Data Fig. 4 and Supplementary Text 10).

To determine whether identified recombination breakpoints are significantly shifted towards the 3' end of the genome, we performed a permutation test comparing the difference between the mean of the distribution of uniformly simulated breakpoints and the mean of the detected breakpoint position distribution in the true set (Supplementary Text 12). We also conducted a change-point analysis using the changepoint R package[31] and fit a Poisson model to the count of recombination prediction interval midpoints. We then computed the mean rate of recombination breakpoints within the intervals on either side of the identified change point to estimate the fold increase in recombination rate in the 3' portion of the genome (Supplementary Text 13). To estimate $R/M$, we found the decrease in parsimony score associated with each detected recombination event as an estimate of $R$. We then calculated $M$ by taking this value and subtracting it from the total number of mutations observed across our entire phylogeny (Supplementary Text 16). $R/M$ is the ratio of these values.

### Reporting summary

Further information on research design is available in the Nature Research Reporting Summary linked to this article.

### Data availability

All data is available in the manuscript or the supplementary materials. Dataset 1 (containing the phylogeny analysed for recombination in this study in Newick format) and dataset 2 (containing a list of descendant samples of recombinant nodes identified through RIPPLES) are available at https://doi.org/10.5281/zenodo.6717378[32].

### Code availability

RIPPLES software is available under the MIT license as part of the UShER package at https://github.com/yatisht/usher. We provide a reproducible Google Cloud Platform workflow for RIPPLES under https://github.com/yatisht/usher/tree/master/scripts/recombination. An archived version of the specific code and workflow used in this study is available from https://doi.org/10.5281/zenodo.6709991(ref. [33]). We distribute RIPPLES with UShER because it uses the same underlying data objects and UShER is required to infer the input MAT. Documentation for RIPPLES and associated utilities can be found at https://usher-wiki.readthedocs.io/en/latest/.

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

**Acknowledgements** We gratefully acknowledge the authors and the laboratories responsible for obtaining the specimens and the submitting laboratories where the genome data were generated and shared via GISAID (Supplementary Table 2)[26], China National Center for Bioinformation (Supplementary Table 3), COVID-19 Genomics UK (COG-UK)[28] (Supplementary Table 4) and the National Center for Biotechnology Information database[27] (Supplementary Table 5), on which this research is based. We thank S. Mollenkamp for assisting with the code development. B.T., J.M. and R.C.-D. were funded by National Institutes of Health grant no. R35GM128932. R.C.-D. was funded by an Alfred P. Sloan Foundation fellowship and the University of California Office of the President Emergency COVID-19 Research Seed Funding Grant no. R00RG2456. B.T. and J.M. were funded by the National Institutes of Health grant no. T32HG008345. B.T. was funded by the National Institutes of Health grant no. F31HG010584. N.D.M. was funded by the European Molecular Biology Laboratory. R.L. was funded by Australian Research Council grant no. DP200103151 and an Chan-Zuckerberg Initiative grant.

Additional funding for this project was provided by Eric and Wendy Schmidt by recommendation of the Schmidt Futures programme.

**Author contributions** R.C.-D. and Y.T. developed the approach and wrote the manuscript. R.C.-D., Y.T., B.T. and R.L. designed experiments. Y.T., B.T., A.H. and N.D.M. conducted experiments. Y.T., B.T., A.H., J.M., N.A., K.S. and C.Y. developed code. R.C.-D. and D.H. supervised the group. Y.T., B.T., A.H., J.M., N.A., C.Y., N.D.M., D.H., R.L. and R.C.-D. edited the manuscript.

**Competing interests** R.L. works as an advisor to GISAID. The remaining authors declare no competing interests.

**Additional information**
**Correspondence and requests for materials** should be addressed to Yatish Turakhia or Russell Corbett-Detig.

A

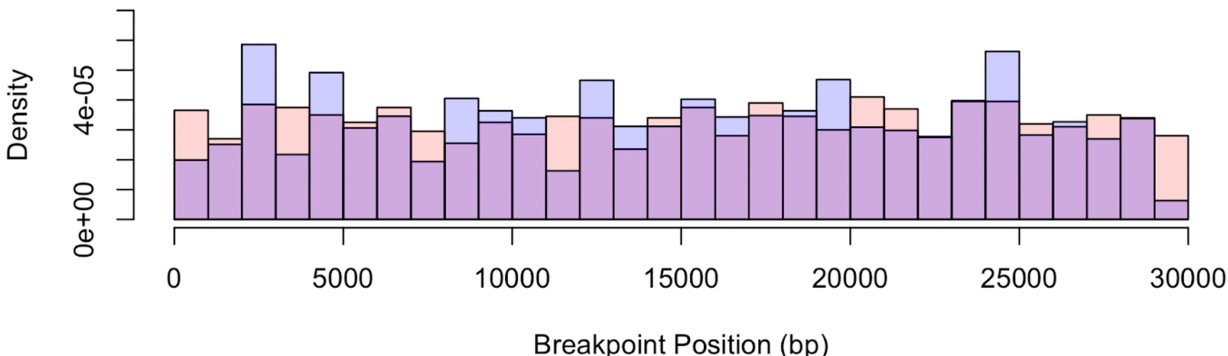

B

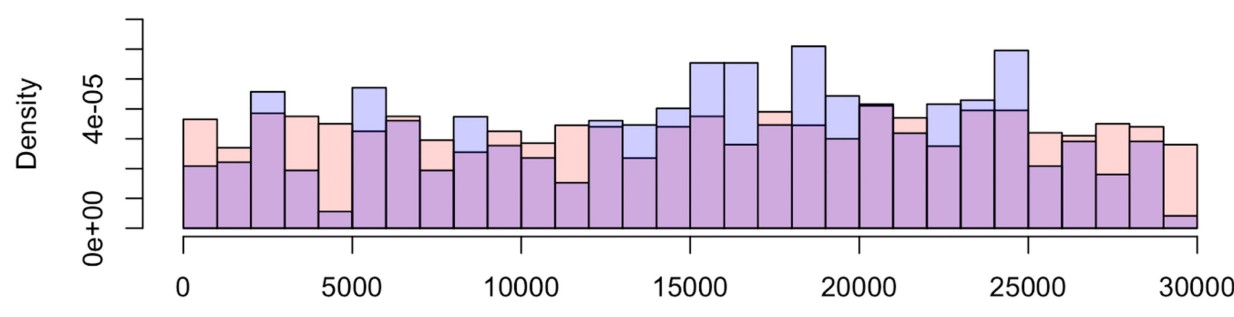

**Extended Data Fig. 1 | Histogram of inferred and simulated recombination breakpoint positions. A**) True simulated breakpoints (red) are shown with all detected recombination interval midpoints (blue). Where blue bars exceed the height of red, it implies an excess rate of detection relative to the true rate of breakpoint positions. Likewise, where red bars exceed the height of blue, it implies a deficit. **B**) True simulated breakpoints (red) are shown with detected recombination interval midpoints for the 20% of the most closely related donor-acceptor pairs (blue). In both comparisons, we broke ties between equivalently improved partial phylogenetic placement parsimony scores by selecting the largest recombination intervals.

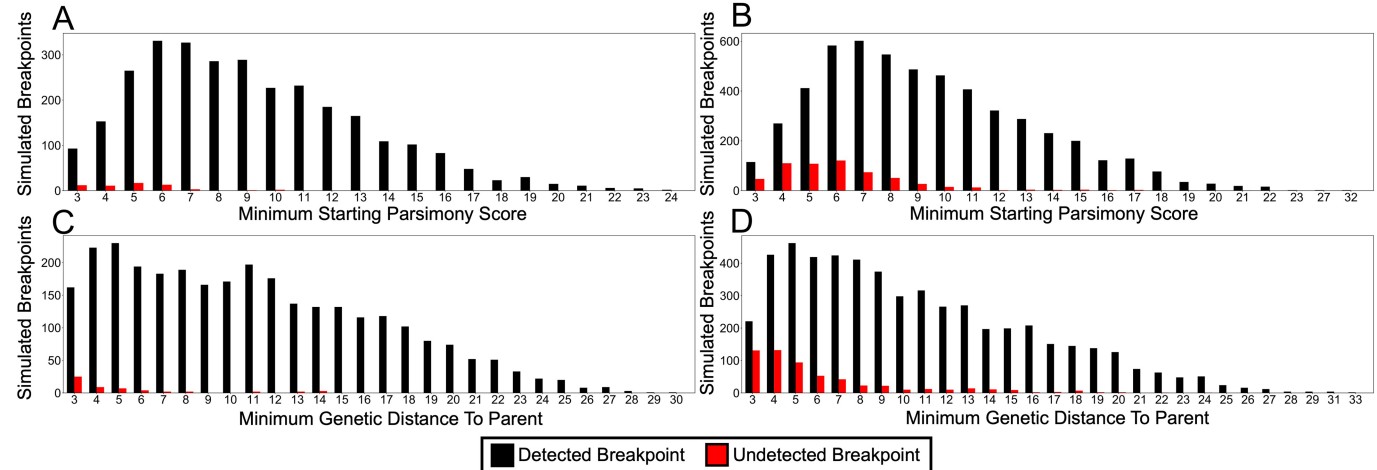

**Extended Data Fig. 2 | RIPPLES more easily detects breakpoints causing large changes in parsimony score.** The distribution of simulated breakpoints detected for each simulated sample is shown for each sample by **A**) initial parsimony score and **B**) minimum genetic distance from simulated sample to parent. Initial parsimony (A) is dependent upon the initial placement of the recombinant node in the tree and refers to the genetic distance in mutations between the recombinant node and its direct parent in the phylogeny. Minimum genetic distance from sample to parent (**B**) refers to the number of mutations relevant to recombination that separate the recombinant node from either the donor or the acceptor, and is not dependent on the initial phylogeny.

Similarly, among the simulated samples detected by RIPPLES, the detected and undetected breakpoints are shown by **C**) initial parsimony score and **D**) minimum genetic distance to parent. Detected samples and breakpoints are shown in black and undetected samples and breakpoints are shown in red. We condition on locating the true breakpoints and observing a significant parsimony score according to our phylogenetic null model. Therefore, we exclude recombination events with minimum starting parsimony scores and genetic distances of less than 3, as these are not significant under our null model.

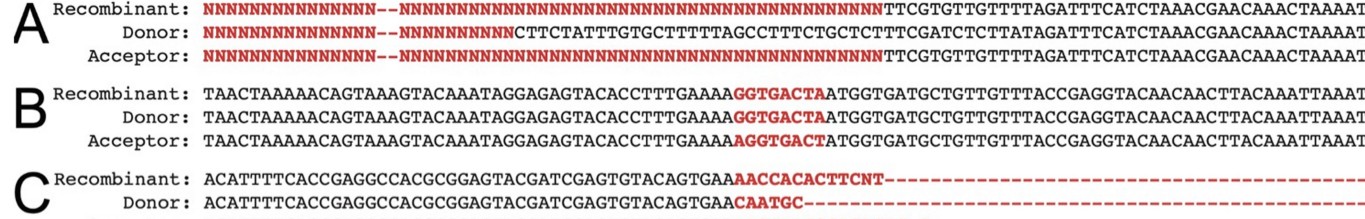

A
```
Recombinant: NNNNNNNNNNNNNNNN--NNNNNNNNNNNNNNNNNNNNNNNNNNNNNNNNNNNNNNNNNNNNNTTCGTGTTGTTTTAGATTTCATCTAAACGAACAAACTAAAAT
     Donor: NNNNNNNNNNNNNNNN--NNNNNNNNNNCTTCTATTTGTGCTTTTTAGCCTTTCTGCTCTTTCGATCTCTTATAGATTTCATCTAAACGAACAAACTAAAAT
  Acceptor: NNNNNNNNNNNNNNNN--NNNNNNNNNNNNNNNNNNNNNNNNNNNNNNNNNNNNNNNNNNNNNTTCGTGTTGTTTTAGATTTCATCTAAACGAACAAACTAAAAT
```

B
```
Recombinant: TAACTAAAAACAGTAAAGTACAAATAGGAGAGTACACCTTTGAAAAGGTGACTAATGGTGATGCTGTTGTTTACCGAGGTACAACAACTTACAAATTAAAT
     Donor: TAACTAAAAACAGTAAAGTACAAATAGGAGAGTACACCTTTGAAAAGGTGACTAATGGTGATGCTGTTGTTTACCGAGGTACAACAACTTACAAATTAAAT
  Acceptor: TAACTAAAAACAGTAAAGTACAAATAGGAGAGTACACCTTTGAAAAAGGTGACTATGGTGATGCTGTTGTTTACCGAGGTACAACAACTTACAAATTAAAT
```

C
```
Recombinant: ACATTTTCACCGAGGCCACGCGGAGTACGATCGAGTGTACAGTGAAAACCACACTTCNT-----------------------------------------
     Donor: ACATTTTCACCGAGGCCACGCGGAGTACGATCGAGTGTACAGTGAACAATGC------------------------------------------------
  Acceptor: ACATTTTCACCGAGGCCACGCGGAGTACGATCGAGTGTACAGTGAAAACCAACCGAGACTG-C-------------------------------------
```

**Extended Data Fig. 3 | Examples of detected trios filtered out due to sequence quality concerns. A**) Partial alignment of consensus sequences from a filtered recombinant trio of nodes 77695, 169585, and 77690, centred on site 28225, has consensus sequences of mostly 'N' spanning several sites meant to be informative of a recombination event. This can occur when many descendant samples have missing data. Mismatches between the three consensus sequences immediately flanking this region may be the result of poor sequencing quality as well. **B**) Partial alignment of consensus sequences from a filtered recombinant trio of nodes 173213, 173209, and 173274, centred on site 16846, has 7 recombination-informative mutations in an 8-nucleotide window that are unlikely to be true mutation events, but rather an alignment artifact or a complex indel event. **C**) Partial alignment of consensus sequences from a filtered recombinant trio of nodes 293461, 293460, and 211841, centred on site 29769, has 3 mismatches in a 5-nucleotide window, immediately flanked by a large gap in the alignment and are unlikely to be true mutations.

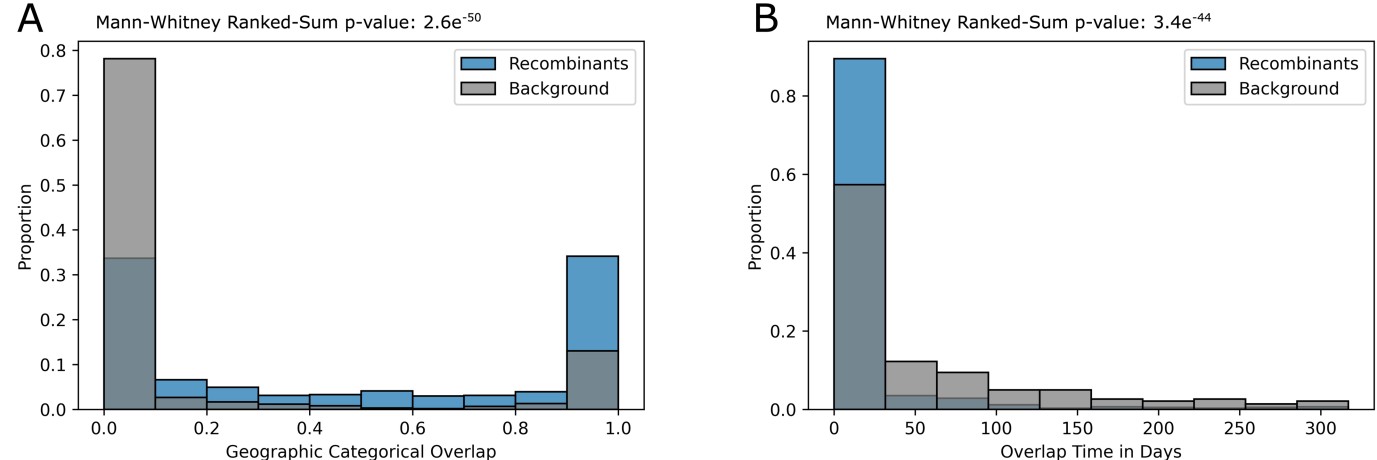

**Extended Data Fig. 4 | Recombinant ancestors exhibit increased spatial and temporal overlap. A)** Spatial and **B)** temporal overlap for our recombinant trios (in blue) and the null distribution (in gray), with Mann-Whitney Ranked-Sum p-values for the statistical increase in overlap for the recombinant ancestors shown on the top.

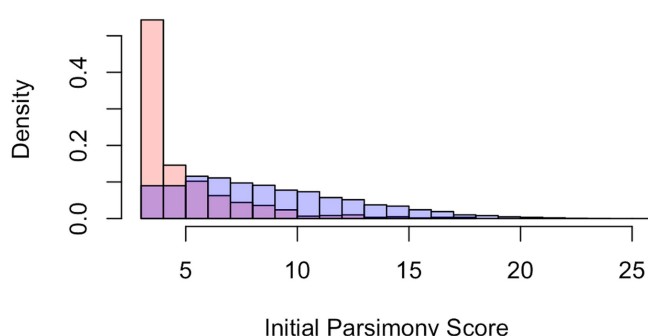

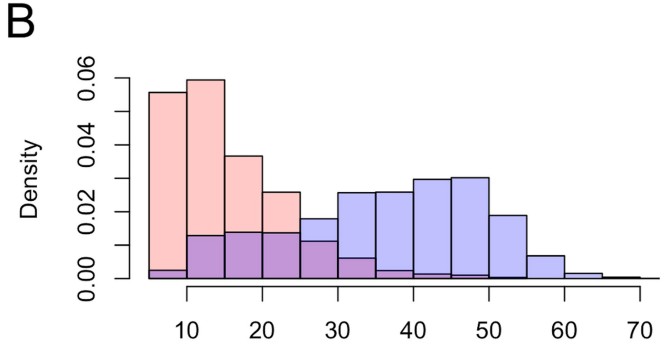

**Extended Data Fig. 5 | Ancestors of recombinants are genetically similar.**
**A**) The initial parsimony scores for placements of putative (red) and simulated
(blue) recombinant samples. **B**) The genetic distance between inferred (red)
and simulated (blue) ancestor-donor pairs that gave rise to putative or
simulated recombinants.

**Extended Data Table 1 | Summary of simulated breakpoint detection**

| Simulation Type | Detected Breakpoints | Total Detectable Breakpoints | Sensitivity |
|---|---|---|---|
| One Breakpoint, No Added Mutations | 196 | 203 | 0.966 |
| One Breakpoint, One Added Mutation | 198 | 204 | 0.971 |
| One Breakpoint, Two Added Mutations | 168 | 179 | 0.939 |
| One Breakpoint, Three Added Mutations | 181 | 191 | 0.948 |
| Two Breakpoints, No Added Mutations | 343 | 384 | 0.893 |
| Two Breakpoints, One Added Mutation | 316 | 360 | 0.878 |
| Two Breakpoints, Two Added Mutations | 340 | 388 | 0.876 |
| Two Breakpoints, Three Added Mutations | 312 | 364 | 0.857 |
| **Total, One Breakpoint** | 743 | 777 | 0.956 |
| **Total, Two Breakpoints** | 1311 | 1496 | 0.876 |
| **Total** | 2054 | 2273 | 0.904 |

If a simulated recombinant had only statistically insignificant parsimony improvements, it is not included here as we consider this recombination event undetectable.

**Extended Data Table 2 | Raw sequence read datasets used to confirm recombination informative positions in selected recombinant samples**

| recombinant_node | Recombinant accession | Sample ID |
|---|---|---|
| 55577 | ERR5860975 | EPI_ISL_722494 |
| 224689 | ERR5433158 | EPI_ISL_1180452 |
| 45828 | ERR5409646 | QEUH-121CC26 |
| 54010 | ERR5064277 | QEUH-A4D8D8 |
| 357644 | ERR4671078 | MILK-991B91 |
| 239616 | ERR5220136 | LOND-1323405 |
| 22683 | ERR5965948 | MILK-1580FB8 |
| 44547 | ERR5070101 | PHWC-490FD7 |
| 88824 | ERR5677159 | QEUH-144D8CC |
| 43018 | ERR5065119 | QEUH-AAF133 |

**Extended Data Table 3 | False discovery rate estimation for each parsimony score improvement observed in our dataset**

| Country | 3'/5' Rate Ratio | P value |
|---------|-----------------|---------|
| USA | 2.94 | <2.2e-16 |
| England | 2.4 | 0.0003944 |
| India | 2.65 | 6.81E-06 |
| Turkey | 1.99 | 0.02286 |
| France | 2.23 | 2.79E-05 |

**Extended Data Table 4 | Increased rate of breakpoint interval midpoint in the 3' portion of the genome when the recombinants are subdivided by the country of origin**

| Country | 3'/5' Rate Ratio | P value |
|---|---|---|
| USA | 2.94 | <2.2e-16 |
| England | 2.4 | 0.0003944 |
| India | 2.65 | 6.81E-06 |
| Turkey | 1.99 | 0.02286 |
| France | 2.23 | 2.79E-05 |

# nature research

Russell Corbett-Detig

# Reporting Summary

Nature Research wishes to improve the reproducibility of the work that we publish. This form provides structure for consistency and transparency in reporting. For further information on Nature Research policies, see our Editorial Policies and the Editorial Policy Checklist.

## Statistics

For all statistical analyses, confirm that the following items are present in the figure legend, table legend, main text, or Methods section.

| n/a | Confirmed | |
|---|---|---|
| ☒ | ☐ | The exact sample size (*n*) for each experimental group/condition, given as a discrete number and unit of measurement |
| ☐ | ☒ | A statement on whether measurements were taken from distinct samples or whether the same sample was measured repeatedly |
| ☐ | ☒ | The statistical test(s) used AND whether they are one- or two-sided *Only common tests should be described solely by name; describe more complex techniques in the Methods section.* |
| ☒ | ☐ | A description of all covariates tested |
| ☐ | ☒ | A description of any assumptions or corrections, such as tests of normality and adjustment for multiple comparisons |
| ☐ | ☒ | A full description of the statistical parameters including central tendency (e.g. means) or other basic estimates (e.g. regression coefficient) AND variation (e.g. standard deviation) or associated estimates of uncertainty (e.g. confidence intervals) |
| ☐ | ☒ | For null hypothesis testing, the test statistic (e.g. *F*, *t*, *r*) with confidence intervals, effect sizes, degrees of freedom and *P* value noted *Give P values as exact values whenever suitable.* |
| ☒ | ☐ | For Bayesian analysis, information on the choice of priors and Markov chain Monte Carlo settings |
| ☒ | ☐ | For hierarchical and complex designs, identification of the appropriate level for tests and full reporting of outcomes |
| ☒ | ☐ | Estimates of effect sizes (e.g. Cohen's *d*, Pearson's *r*), indicating how they were calculated |

*Our web collection on statistics for biologists contains articles on many of the points above.*

## Software and code

Policy information about availability of computer code

| | |
|---|---|
| Data collection | All data used in this work are available from GISAID (gisaid.org), COG-UK, and Genbank, with specific sample accessions listed in Supplemental Tables 5-8. |
| Data analysis | The data was analyzed using code available at https://github.com/yatisht/usher and https://github.com/bpt26/recombination. All software versions are indicated where appropriate in the methods section of the manuscript. |

For manuscripts utilizing custom algorithms or software that are central to the research but not yet described in published literature, software must be made available to editors and reviewers. We strongly encourage code deposition in a community repository (e.g. GitHub). See the Nature Research guidelines for submitting code & software for further information.

## Data

Policy information about availability of data

All manuscripts must include a data availability statement. This statement should provide the following information, where applicable:
- Accession codes, unique identifiers, or web links for publicly available datasets
- A list of figures that have associated raw data
- A description of any restrictions on data availability

All data used in this work are available from GISAID (gisaid.org), COG-UK, and GenBank, with specific sample accessions listed in Supplemental Tables 5-8.

# Field-specific reporting

Please select the one below that is the best fit for your research. If you are not sure, read the appropriate sections before making your selection.

☐ Life sciences ☐ Behavioural & social sciences ☒ Ecological, evolutionary & environmental sciences

For a reference copy of the document with all sections, see nature.com/documents/nr-reporting-summary-flat.pdf

# Ecological, evolutionary & environmental sciences study design

All studies must disclose on these points even when the disclosure is negative.

| | |
|---|---|
| Study description | In this study, we describe an efficient method that exhaustively searches a phylogeny with applications demonstrated for the current SARS-CoV-2 global phylogeny. We compared our approach to many existing methods and documented accuracy (on simulated data), consistency (with empirical data), compute time and memory usage requirements. |
| Research sample | Our study is based on existing dataset of SARS-CoV-2 sequences shared via GISAID (gisaid.org), GenBank, and COG-UK. The specific sample accessions are listed in Supplementary Tables 5-8. |
| Sampling strategy | Not relevant. We chose to work primarily with our 28/5/21 public release of the SARS-CoV-2 phylogeny, because in order to develop our software, we needed a constant tree to perform experiments on and these were the most up-to-date available at the time we began this work. We also worked with simulated data, designed to behave similarly to the real data, as described in our Methods section. |
| Data collection | All sequences marked as 'complete' and 'high coverage' submitted up to 28/5/21 were downloaded from GISAID (gisaid.org), as well as sequences from GenBank, and COG-UK, were used to build the global phylogeny after a few additional filtering steps (Methods). These data are from a collection of sequences obtained throughout the world during the SARS-CoV-2 pandemic. Supplementary Tables 5-8 list all individuals responsible for the primary data collection in all sequences used in this study. |
| Timing and spatial scale | All sequences present in the 28/5/2021 public tree were used, except for those pruned out according to our Methods section. We chose 28/5/21 because we needed a consistent sample with which to hone our methods and conduct experiments, as well as to have a "reference tree" to refer back to throughout the study. |
| Data exclusions | Incomplete and low-coverage sequences as well as those with known sequence issues were excluded (Methods). Our previous study and other related studies cited in the Methods demonstrate that errors can lead to false nucleotide substitutions for myriad reasons unrelated to the biology of the virus itself. We have masked these sites from our analysis and the specific criteria for exclusion are indicated in the method section. |
| Reproducibility | All our findings and results are completely reproducible using the code and data available from https://github.com/yatisht/usher. Simulations and filtration of sequences were conducted using code from https://github.com/bpt26/recombination. |
| Randomization | Not relevant. We used identical dataset for all comparative analysis hence randomization is not necessary for comparing results of the approaches used in this study. |
| Blinding | Blinding is not relevant because experimenter bias cannot affect the results of this analysis. |

Did the study involve field work? ☐ Yes ☒ No

# Reporting for specific materials, systems and methods

We require information from authors about some types of materials, experimental systems and methods used in many studies. Here, indicate whether each material, system or method listed is relevant to your study. If you are not sure if a list item applies to your research, read the appropriate section before selecting a response.

## Materials & experimental systems

| n/a | Involved in the study |
|---|---|
| ☒ | Antibodies |
| ☒ | Eukaryotic cell lines |
| ☒ | Palaeontology and archaeology |
| ☒ | Animals and other organisms |
| ☒ | Human research participants |
| ☒ | Clinical data |
| ☒ | Dual use research of concern |

## Methods

| n/a | Involved in the study |
|---|---|
| ☒ | ChIP-seq |
| ☒ | Flow cytometry |
| ☒ | MRI-based neuroimaging |

