## [Peer Review File · Nature]

Manuscript Title: Pandemic-Scale Phylogenomics Reveals The SARS-CoV-2 Recombination Landscape

Reviewer Comments & Author Rebuttals

Reviewer Reports on the Initial Version:

Referees' comments:

Referee #1 (Remarks to the Author):

Summary of the key results

Turkahia et al. present a novel method for the detection of recombination based on maximum parsimony, which they apply to a SARS-CoV-2 dataset that represents the majority of genomes sequenced globally to date. The method's statistical properties are assessed by simulation. From the application of their method to the data, they draw conclusions about the prevalence of detectable recombination in SARS-CoV-2, and about the genomic distribution of breakpoints. More generally, the method is presented as a solution to the problem of conducting pandemic-scale analyses at a time when the scale of genomic sequencing means that some/many previously-developed analysis methods have become infeasible.

Originality and significance

The method is a timely and potentially significant development in the field and to the best of my knowledge is novel; its application at this scale certainly is. The finding of recombination in SARS-CoV-2 is not novel, as the authors note, it has been demonstrated to occur by Jackson et al. (2021). I have some concerns about the main biological conclusions drawn by the study (see below).

Data & methodology

The data, which are a QCed 1.6million tip mutation-annotated maximum parsimony tree representing most of the pandemic to May, are an appropriate dataset available to test the method and the authors' biological hypotheses. One thing that did occur to me was whether the fact that recombination is being inferred using a phylogeny which is itself affected by recombination was likely to affect the results in any way. Have the authors considered this?

Conclusions: robustness, validity, reliability

I think that the authors present two main biological conclusions in this work, firstly the genomic distribution of recombination breakpoints, and secondly the overall prevalence of recombination. I

will address them in turn.

In figure 2A the authors present the finding that there is an excess of breakpoints detected in the 3' region of the genome, and suggest that this is evidence for an excess of recombination in spike. I am concerned that this finding might be an artefact of variation segregating at a higher frequency at the 3' end of the genome compared to in the polymerase gene. I attach a plot which is the density of non-reference nucleotide counts along the genome in 500,000 randomly chosen genomes out of the 2.8 million available on gisaid at the time of writing. I can also find a positive relationship between genomic position and minor allele frequency in the same data. My guess would be that SARS-CoV-2 is less constrained at the 3' end than at the 5' end of the genome, so that segregating variation at the 3' end is more likely to reach higher frequency. Because the detection of recombination is dependent on genetic diversity, I think this pattern could potentially explain at least some of the relationship in their figure 2A. If I have understood the authors' simulation procedure correctly, they control for the effective mutation rate across the genome, but do not control for different levels of selective constraint (allele frequency)? I think this needs to be addressed to rule out the possibility that the patterns they find are due to higher power to detect recombination in regions with higher genetic diversity.

The authors identify 606 recombination events and state that these are ancestral to 2.7% of sequenced SARS-CoV-2 genomes. They are careful to explain that this figure is for detectable recombination events and the true amount of recombination is likely to be much greater. I agree with these points, but given this, it leaves me wondering how interesting the figure of 2.7% actually is, beyond its relevance to the problem of building accurate phylogenies. Most template switching by the polymerase during genome replication is likely occurring between genetically identical viral RNA in the case of infection from a single virus, and I would guess that most true co-infections are probably between genetically similar viruses which are circulating in the same locality.

Suggested improvements: experiments, data for possible revision

It would be nice to see the simulation study extended to incorporate possible differences in levels of genetic diversity across the genome as above, to rule out the possibility that the pattern in figure 2A is due to ascertainment bias. I would also be interested in knowing to what extent, if any, the method depends on the "correctness" of the input tree.

Clarity and context: lucidity of abstract/summary, appropriateness of abstract, introduction and conclusions

The context and the clarity are good throughout the manuscript. The abstract presents the main points of the paper succinctly.

Regarding the sentence: "During the SARS-CoV-2 pandemic, genomic data generation has exceeded the capacities of existing analysis platforms, thereby crippling real-time analysis of viral recombination 5" - reference 5 does not mention recombination; this citation is appropriate for the

first part of the sentence, but not the last.

I like the method that the authors present and am excited to see it used in the future. I would just note that 1) downsampling (“reducing the search space for possible pairs of recombinant ancestors”), which the authors contrast to their approach at points in the manuscript, is not an inappropriate procedure given that there is a lot of redundancy in the SARS-CoV-2 data and one need only look for recombination between viruses which are in the same place at the same time. 2) Ripples’ run time of four days using 225 CPUs is itself not insignificant. Will that run time increase with the number of nodes in the tree as well as with the amount of genetic diversity in the putative children (i.e. not linearly) as the pandemic progresses?

References

Jackson, Ben, Maciej F. Boni, Matthew J. Bull, Amy Colleran, Rachel M. Colquhoun, Alistair C. Darby, Sam Haldenby, et al. 2021. “Generation and Transmission of Inter-Lineage Recombinants in the SARS-CoV-2 Pandemic.” *Cell*, August. <https://doi.org/10.1016/j.cell.2021.08.014>.

Referee #2 (Remarks to the Author):

The Authors present a novel method to detect recombination in SARS-CoV-2 genomes. This addresses a challenging problem because of the scale which such data has become available and because of the relatively limited sequence divergence, which may not offer strong support for recombination, in particular in the presence of recurrent substitutions that likely occurred through directional selection. There is a clear need to go beyond the description of anecdotal or specific obvious recombinants and to perform a comprehensive systematic assessment. For this purpose, the authors developed a sensitive phylogenetic approach, as assessed through simulations, and make considerable efforts to avoid false positives.

While I believe the study offers a very useful quantitative assessment of measurable recombination signal in SARS-CoV-2, I am not fully convinced about the finding of a higher degree of recombination in the Spike gene. This is possible and could indeed be in line with recombination patterns detected in other coronaviruses. However, the spike gene is subject to considerable diversifying selection resulting in pronounced standing diversity. It also experiences considerable directional selection resulting in many recurring mutations (e.g. <https://doi.org/10.1016/j.cell.2021.09.003>). As a consequence, it may be easier to detect recombination in spike and it may also be more prone to false positives. The latter may be to a large extent avoided by the way the null model is set up, but I am more concerned about the former. The Authors try to address the support in the excess of 3' recombination using a permutation test, but I am not convinced this removes a possible detection bias in the real data. An approach with 93% sensitivity may almost invariably pick up uniformly ad random simulated recombinants (simulated using the approach used here), but in the real data more recombination signal may still exceed the detection threshold in more variable regions. I believe other simulation procedures are needed to address this, e.g. by uniformly simulating recombination across the genome with its inherent spatial variability, allowing a homogeneous range of outcomes from no detectable recombination imprint to a strong imprint and assessing

whether there is a uniform detection rate despite the genome variability pattern.

There is a wide range in false discovery rates across events associated with a specific starting parsimony score and improvement (Table 4). This makes me wonder if this should not be taken into account in the recombination identification. For example, for recombination events with a starting parsimony score of 6 and an improvement of 6, the FDR is > 0.5 . Can we ignore this and only consider the average FDR of 0.13?

As the Authors point out, recombination detection in genetically similar viral lineages is challenging and will lead to underestimates of the overall frequency of recombination. It therefore seems remarkable that about 40% of the events are detected within Pango lineages. It is possible that this reflects for example more recombination opportunity due higher probability of co-circulation. Some lineages like B.1.1.7 may also be densely sampled offering a lot of opportunity to pick up recombinants. I think it might be useful to plot the frequency distribution of recombination with respect to the genetic distance between the donor and acceptor lineages.

For the practitioner, it would be useful to be able to check whether data sets include any genomes that are descendants of the recombination events detected here (and filter them). This would require making available a list of the 43,163 descendant samples (and perhaps updating this if the Authors are doing real-time detection).

Minor comments

"Additionally, excess similarity of geographic location and date metadata among the descendants of donor and acceptor nodes strongly supports the notion that the ancestors of recombinant genomes co-circulated within human populations (Supplementary Text 10, Supplementary Fig. 4) – a prerequisite for recombination."

This seems to be an overstatement -- there is a significant difference in overlap for the identified recombinants, but no strong evidence that there was co-circulation for all recombinants. There is still a considerable fraction for which there is no detectable spatial overlap for example (Fig. S4A)

"Furthermore, a recombination event that transferred a portion of the Spike protein coding region into the ancestor of SARS-CoV-2 may have contributed to the emergence of the COVID-19 pandemic in human populations 11."

This has been shown to be a misinterpretation due to the fact that the bat RaTG13 virus is significantly more divergent in the receptor binding and thus the plausible recombinant (<https://doi.org/10.1038/s41564-020-0771-4>), as also independently confirmed by (<https://doi.org/10.1093/ve/veaa098>).

SI: "As would be expected, more modest parsimony score improvements are associated with a higher estimated false discovery rate (Supplementary Table 3)." Should this not refer to Table 4?

Referee #3 (Remarks to the Author):

There have been a number of recent papers and preprints (cited by the authors) investigating recombination within Sars COV-2, which are broadly consistent with each other.

This analysis presents a new method and the biggest survey of recombination to date based on looking for signals within a phylogeny generated using 1.6 million genomes. The signal used by the method to find putative recombination is long branch length in the reconstructed phylogenetic trees. For these lineages, the method makes exhaustive search of putative hybridisation events. The method works well in highly sampled organism but would likely be less reliably in poorly sampled organisms (which are the great majority) because recombination only reliably increases branch lengths in the presence of intermediate sequences. If these are absent other signals may be more effective. Since the paper claims to present a general new methodology, this issue of when it works well does need to be addressed.

Furthermore, the test of the method is mainly a test of whether it can identify manually generated recombination events, with a very optimistic headline figure of 93% detection rate is given. For many real world datasets (less so for COV-2 because of the density of sampling) the question of whether the position of the donor sequences in the phylogeny was accurately estimated would be very important but this aspect is not tested at all.

This work really makes two claims for newsworthiness. The first is that the method is easy enough to use that it can become a routine part of analysis of viral sequence. I think this is valid in terms of useability, but with the proviso that it has not been proven to be applicable to less densely sampled organisms as described above.

The second is to be a comprehensive analysis of recombination within Sars COV2. For example, they find 600 putative recombination events, while a recent paper of Varabyue published in Genetics finds 200. They make the specific claim that around 2.7% of SARS-Cov2 sequences are recombinant and estimate that only a very small fraction of genetic changes are currently made by recombination. I basically buy the validity of these analyses and the statistics generated are novel and are likely to be well cited.

Nothing is said about the distribution of recombination events on the tree, e.g. in terms of age distribution and frequency in particular lineages. Geographical information would also be very interesting.

The manuscript also finds that recombination is higher in the last part of the gene, where the spike protein is and finds that unsurprisingly but reassuringly recombination occurs preferentially between lineages that are found to co-occur in time and space.

The paper also claims that this is due to mechanistic factors rather than natural selection. While mechanistic factors are perfectly plausible, I don't find the arguments for this especially convincing. The selection responsible may very well be transitory in nature, e.g. due to immune selection in individual hosts and the fact that the lineages are not obviously successful is poor evidence for the absence of selection.

The manuscript claims in the abstract that identifying recombination events is important from a practical point of view essentially to identify variants of concern, but I think this is very much unproven at least in the context of evolution within SARS-coV2. Probably the lineages most likely to be VOC are those where multiple substitutions have arisen de novo, driven by selection (and epistasis). There is no evidence that any of the existing VOCs arose by recombination.

Referee #4 (Remarks to the Author):

This is an interesting paper on the role of recombination in the on going pandemic. The authors perform a very exhaustive recombination analysis, involving more than one million sequences available worldwide. For this purpose, they developed a novel method (RIPPLES) which leverages the information from an input (large) phylogenetic tree to quickly detect recombination events.

In my opinion, the novelty of this work is based on the detection of hundreds of recombination events in SARS-CoV-2, rather than on the results or approach shown here. While previous works have reported recombination previously (e.g. Jackson et al, 2021), these studies were focused on much smaller datasets and/or specific viral lineages. However, some of the results highlighted in the manuscript are either already known (e.g. unequal distribution of recombination breakpoints in Coronaviruses have been reported in several publications) or have an unclear impact in terms of clinical or biological relevance (e.g. B.1.355 lineage being a recombinant).

In addition, I have the following specific major comments:

- It is noteworthy that one of the problems when detecting recombination is the lack of power to detect it at low levels of genetic diversity. Indeed, recombination between closely related sequences does not necessarily increase diversity/branch lengths. This method does not address this limitation. It only focuses on recombination events that cause the generation of branches of longer length. Thus, the detection of recombination events may be skewed towards those events occurring between relatively distant parental lineages.
- Sup Figures 1 and 2 show some plots on the performance of RIPPLES. I cannot find the study showing a rate of false positives, and negatives, under different scenarios of i) genetic variability, ii) recombination rate (expected number of recombination breakpoints found in the simulation), iii) sample size of the dataset. Very importantly, this work lacks a comparison with pre-existing methods for the detection of recombination (RDP, 3seq, Bootscan, etc).
- Finally, the authors claim that this method is very fast. However, I understand that this was calculated from a starting point that includes a precomputed phylogenetic tree as input. Thus, we need to add the time of tree inference to computational time? Also, these computational times have been measured running RIPPLES in really large computers (224 vCPUs). Would it be possible to analyze these SARS-CoV-2 samples in a smaller computer?

Minor comments:

- Authors should specify clearly how they define genetic distance. Number of nucleotide differences?

Did they use any specific substitution model?

- Regarding the performance: Is the false discovery rate of 11.6% affected by any of the post hoc filters mentioned in the manuscript?

Author Rebuttals to Initial Comments:

Referees' comments:

Referee #1 (Remarks to the Author):

Summary of the key results

Turkahia et al. present a novel method for the detection of recombination based on maximum parsimony, which they apply to a SARS-CoV-2 dataset that represents the majority of genomes sequenced globally to date. The method's statistical properties are assessed by simulation. From the application of their method to the data, they draw conclusions about the prevalence of detectable recombination in SARS-CoV-2, and about the genomic distribution of breakpoints. More generally, the method is presented as a solution to the problem of conducting pandemic-scale analyses at a time when the scale of genomic sequencing means that some/many previously-developed analysis methods have become infeasible.

Originality and significance

The method is a timely and potentially significant development in the field and to the best of my knowledge is novel; its application at this scale certainly is. The finding of recombination in SARS-CoV-2 is not novel, as the authors note, it has been demonstrated to occur by Jackson et al. (2021). I have some concerns about the main biological conclusions drawn by the study (see below).

We thank the reviewer for the positive feedback. We agree with the reviewer that our method for recombination inference and its scalability is truly novel and the most significant component of our work. We have revised the manuscript title taking your feedback into consideration.

Data & methodology

The data, which are a QCed 1.6million tip mutation-annotated maximum parsimony tree representing most of the pandemic to May, are an appropriate dataset available to test the method

and the authors' biological hypotheses. One thing that did occur to me was whether the fact that recombination is being inferred using a phylogeny which is itself affected by recombination was likely to affect the results in any way. Have the authors considered this?

We have considered this possibility. Indeed, discrepancies that result from a single-tree representation of the data and the underlying network imposed by recombination is at the core of our method. Specifically, we examine relatively long branches (defined as those with ≥ 3 total mutations) to prune the search space when looking for recombination. To solve for "tree correctness", we revised our simulation strategy to search for recombination on a much smaller dataset. Specifically, we selected 1,000 random samples each from two large clades, and then created a phylogeny from scratch using UShER and matOptimize. The rationale behind this is that smaller trees are much less likely to contain errors, while still providing enough phylogenetic context to assess RIPPLES. We then added 500 simulated recombinants to this tree, separately, and ran RIPPLES on the 500 2,001-sample phylogenies, specifically searching for the simulated recombinant. The simulations here provide a "ground truth". Overall, RIPPLES detects 410 (82%) of the 500 simulated breakpoints – a very similar rate of detection to our previous simulation experiments. We detail these results in Text S6.

Conclusions: robustness, validity, reliability

I think that the authors present two main biological conclusions in this work, firstly the genomic distribution of recombination breakpoints, and secondly the overall prevalence of recombination. I will address them in turn.

In figure 2A the authors present the finding that there is an excess of breakpoints detected in the 3' region of the genome, and suggest that this is evidence for an excess of recombination in spike. I am concerned that this finding might be an artefact of variation segregating at a higher frequency at the 3' end of the genome compared to in the polymerase gene. I attach a plot which is the density of non-reference nucleotide counts along the genome in 500,000 randomly chosen genomes out of the 2.8 million available on gisaid at the time of writing. I can also find a positive relationship between genomic position and minor allele frequency in the same data. My guess would be that SARS-CoV-2 is less constrained at the 3' end than at the 5' end of the genome, so that segregating variation at the 3' end is more likely to reach higher frequency. Because the detection of recombination is dependent on genetic diversity, I think this pattern could potentially explain at least some of the relationship in their figure 2A. If I have understood the authors' simulation procedure correctly, they control for the effective mutation rate across the genome, but do not control for different levels of selective constraint (allele frequency)? I think this needs to be addressed to rule out the possibility

that the patterns they find are due to higher power to detect recombination in regions with higher genetic diversity.

Please note that all reviewers raised similar concerns. Here we address these considerations collectively for completeness and clarity of presentation. Below we attempt to address more specific concerns that each reviewer raises.

There are two somewhat distinct considerations that might affect the spatial distribution of discovered recombination events. The first is the possibility of an increased rate of false positives owing to convergent mutations in the spike protein region (see reviewer 2). Reviewer 1 and 2 provide a clear description of our null model. In the null model we define, mutations are introduced as a means of evaluating the impact of additional mutation that might occur on recombinant lineages that were unsampled for some time. As such, we expect that these mutations would be introduced at a rate proportional to their parsimony score. That is, we are explicitly assuming a mutational process which should mitigate some of the concern about elevated rates of false positives. We agree with reviewer 2 that this is unlikely to be a major concern. There are two reasons. First, a false signature of recombination would require a few restrictive circumstances. It requires that a very specific set of mutations evolve convergently, but also that they evolve on a particularly long branch with few intermediates. Although this has clearly occurred it is extremely rare. Second, empirically, when we identify putative recombination events between VOC and non-VOC lineages, VOC spike amino acid mutations are, in fact, slightly less likely to be included in the descendant genome. This is the opposite of the prediction for a higher rate of false positives associated with recurrent mutations.

The second consideration is that statistical power might vary across the SARS-CoV-2 genome (all reviewers touched on this consideration in some ways). In particular, if power is greatest in the spike protein region, we might reasonably expect that the apparent excess is discovered as an artefact. We now present an analysis of the rates of true positives across the SARS-CoV-2 genome in our simulated data (Fig. S1). Our key finding, that we now emphasize more clearly, is that power is reduced towards the extreme 5' and 3' edges of the genome. The concern reflects the fact that RIPPLES is based on detecting consecutive stretches of potentially recombinant mutations. Outside of this bias, which we mitigate in downstream analyses by excluding the first and last 1,000 bp of the viral genome, the true positive rate appears to vary relatively little across the genome when considering the full dataset of simulated recombinants.

Importantly, when we subset the simulated recombinants between the most closely related donor and acceptor nodes (which more closely resembles the set of detected recombinants), we find that the true positive rate varies relatively little. Indeed, the slight bias observed is opposite of the primary pattern we describe where detection within the 5' end of the genome is slightly reduced relative to the center of the virus genome. However, as expected, the true positive rate for

detecting recombinants is reduced across the genome in this subset. We therefore conclude that the pattern of a nearly 3-fold excess of recombination towards the 3' end of the genome is unlikely to result from biases associated with RIPPLES.

These analyses are included in Text S13, S15 and displayed in Fig. S1.

However, we caution that it is not possible to simulate completely biologically accurate recombination events. If additional biases beyond the subsets we considered here affect the distribution of real recombination events in complex ways, this could impact RIPPLES' ability to obtain an unbiased estimate of the recombination events. We have added this consideration to the main text in our manuscript (Page 4).

The authors identify 606 recombination events and state that these are ancestral to 2.7% of sequenced SARS-CoV-2 genomes. They are careful to explain that this figure is for detectable recombination events and the true amount of recombination is likely to be much greater. I agree with these points, but given this, it leaves me wondering how interesting the figure of 2.7% actually is, beyond its relevance to the problem of building accurate phylogenies. Most template switching by the polymerase during genome replication is likely occurring between genetically identical viral RNA in the case of infection from a single virus, and I would guess that most true co-infections are probably between genetically similar viruses which are circulating in the same locality.

Ultimately, all analyses of the relative frequencies of recombination in population and evolutionary genomics depend to some extent on the detectability of recombination events. Recombination is still widely regarded as a potential contributor to the creation of phenotypically novel lineages and should therefore be a major analysis target during the pandemic and into the future of SARS-CoV-2 evolutionary genomics. The demonstration that recombination is present, but has played only a minor role in shaping genetic diversity thus far is crucial for understanding SARS-CoV-2 evolutionary dynamics.

In the context of the pandemic and public health monitoring, detectable recombination is also the recombination that is of more significance. This is because detectable recombination events are more likely to occur between genetically divergent lineages and produce phenotypically novel lineages. That is, recombination events that include many mutations are expected to have the greatest potential to produce "fitness leaps" because many mutations are transposed between genomic backgrounds simultaneously.

Furthermore, while so far SARS-CoV-2 has spread by waves, which has minimized the appearance of important and detectable recombinants, in the future the accumulation of diversity within omicron and the more uncontrolled spread of the virus will give more opportunities for detectable recombinants to appear and to have biological significance. Furthermore, and more importantly, the methods presented here do not apply only to SARS-CoV-2, and as genomic epidemiology becomes more generally adopted, the possible applications of the proposed methods will increase, and will include pathogens where recombination is known to have more impact, for example drug resistance in bacteria or immunity escape in influenza.

Having a framework in place that can detect such events with speed and precision is foundational to modern viral surveillance. RIPPLES is therefore poised to play a central role in the rapid evaluation of newly-produced genome sequences.

Suggested improvements: experiments, data for possible revision

It would be nice to see the simulation study extended to incorporate possible differences in levels of genetic diversity across the genome as above, to rule out the possibility that the pattern in figure 2A is due to ascertainment bias.

In our response to the concern raised above, we have subset the simulated dataset to evaluate not only randomly selected donor/acceptor pairs but also to focus specifically on the most closely related subset of possible donor/acceptor pairs. We believe the response addresses this concern as well. We would also like to emphasize that our simulations start from using real genome sequences (or reconstructed sequences for ancestral nodes) and hence our simulations are taking some of these biases into account.

I would also be interested in knowing to what extent, if any, the method depends on the “correctness” of the input tree.

We agree with the reviewer that tree correctness is a concern, and we do now consider this as a possibility. We revised our simulation strategy to search for recombination on a much smaller dataset. Specifically, we selected 1,000 random samples each from two large clades, and then created a phylogeny from scratch using UShER and matOptimize. The rationale behind this is that smaller trees are much less likely to contain errors, while still providing enough phylogenetic context to assess RIPPLES. We then added 500 simulated recombinants to this tree, separately, and ran RIPPLES on the 500 2,001-sample phylogenies, specifically searching for the simulated recombinant. The simulations here provide a “ground truth”. Overall, RIPPLES detects 410 (82%) of

the 500 simulated breakpoints – a very similar rate of detection to our previous simulation experiments. We detail these results in Text S6.

Clarity and context: lucidity of abstract/summary, appropriateness of abstract, introduction and conclusions

The context and the clarity are good throughout the manuscript. The abstract presents the main points of the paper succinctly.

Regarding the sentence: “During the SARS-CoV-2 pandemic, genomic data generation has exceeded the capacities of existing analysis platforms, thereby crippling real-time analysis of viral recombination 5” - reference 5 does not mention recombination; this citation is appropriate for the first part of the sentence, but not the last.

We have corrected this by revising the statement to state “... crippling real-time analysis of viral evolution [5]”. Recombination dynamics are a subset of evolutionary dynamics so we believe this addresses the point without going into too fine of detail for the manuscript.

I like the method that the authors present and am excited to see it used in the future. I would just note that 1) downsampling (“reducing the search space for possible pairs of recombinant ancestors”), which the authors contrast to their approach at points in the manuscript, is not an inappropriate procedure given that there is a lot of redundancy in the SARS-CoV-2 data and one need only look for recombination between viruses which are in the same place at the same time. 2) Ripples’ run time of four days using 225 CPUs is itself not insignificant. Will that run time increase with the number of nodes in the tree as well as with the amount of genetic diversity in the putative children (i.e. not linearly) as the pandemic progresses?

We thank the reviewer for their valuable insights.

- 1) The issue with downsampling is that it risks reducing the power and accuracy of recombination detection, since it makes branch lengths longer, creating more artefactual possible recombinations (false positives), and reducing the number of donor-acceptor pairs available to test recombination (possibly increasing false negatives). These problems might be exacerbated as diversity increases with the progression of the pandemic and in other datasets with more diversity.**

The runtime will increase during the pandemic. However, because the analysis enabled by RIPPLES is done on a nearly comprehensive phylogenetic tree it need only be done on each major tree updated. After that, RIPPLES is able to search a narrow range of specified samples for potentially recombinant ancestry. In fact, RIPPLES is sufficiently compute/memory efficient that it is possible for a user to add a handful of new samples to the public tree and then search only those samples for recombinant ancestry. On a basic laptop, this takes a few minutes. Text S14 addresses this consideration in detail. We also highlight in Text S18 that inferring recombination at such massive scale was just infeasible with prior techniques.

Besides that, we are consistently working on optimizing and improving the runtime of RIPPLES. In our latest version (UShER release v0.5.3), we have introduced RIPPLES-fast that incorporates several code and data structure optimization techniques to produce the similar output (slight differences arising due to the non-determinism of our previous parallelization approach) as before but is >100-fold faster than the earlier version, requiring only 40 minutes with 224 CPUs for the 1.6M SARS-CoV-2 tree. Post publication, we plan to periodically (daily to weekly) release our RIPPLES recombinant list alongside (and based on) our daily-updated comprehensive SARS-CoV-2 trees (http://hgdownload.soe.ucsc.edu/goldenPath/wuhCor1/UShER_SARS-CoV-2/).

References

Jackson, Ben, Maciej F. Boni, Matthew J. Bull, Amy Colleran, Rachel M. Colquhoun, Alistair C. Darby, Sam Haldenby, et al. 2021. "Generation and Transmission of Inter-Lineage Recombinants in the SARS-CoV-2 Pandemic." *Cell*, August. <https://doi.org/10.1016/j.cell.2021.08.014>.

Referee #2 (Remarks to the Author):

The Authors present a novel method to detect recombination in SARS-CoV-2 genomes. This addresses a challenging problem because of the scale which such data has become available and because of the relatively limited sequence divergence, which may not offer strong support for recombination, in particular in the presence of recurrent substitutions that likely occurred through directional selection. There is a clear need to go beyond the description of anecdotal or specific obvious recombinants and to perform a comprehensive systematic assessment. For this purpose, the authors developed a sensitive phylogenetic approach, as assessed through simulations, and make considerable efforts to avoid false positives.

While I believe the study offers a very useful quantitative assessment of measurable recombination signal in SARS-CoV-2, I am not fully convinced about the finding of a higher degree of recombination

in the Spike gene. This is possible and could indeed be in line with recombination patterns detected in other coronaviruses. However, the spike gene is subject to considerable diversifying selection resulting in pronounced standing diversity. It also experiences considerable directional selection resulting in many recurring mutations (e.g. <https://doi.org/10.1016/j.cell.2021.09.003>). As a consequence, it may be easier to detect recombination in spike and it may also be more prone to false positives. The latter may be to a large extent avoided by the way the null model is set up, but I am more concerned about the former. The Authors try to address the support in the excess of 3'recombination using a permutation test, but I am not convinced this removes a possible detection bias in the real data. An approach with 93% sensitivity may almost invariably pick up uniformly ad random simulated recombinants (simulated using the approach used here), but in the real data more recombination signal may still exceed the detection threshold in more variable regions. I believe other simulation procedures are needed to address this, e.g. by uniformly simulating recombination across the genome with its inherent spatial variability, allowing a homogeneous range of outcomes from no detectable recombination imprint to a strong imprint and assessing whether there is a uniform detection rate despite the genome variability pattern.

We thank the reviewer for identifying a central concern for our analyses. Reviewers 1, 3 and 4, raise similar considerations and we provide a complete response in our response to reviewer 1 above. To briefly address unique concerns that reviewer 2 identifies.

The increase of potential false positives in the spike region is not borne out in simulated data. There are many possible reasons. Two considerations are most pertinent (summarized from above). First, an excess of false positives associated with convergent mutations requires that a very specific set of mutations evolve recurrently, but also that they evolve on a particularly long branch with few intermediates. This appears relatively uncommon. Second, empirically, when we identify putative recombination events between VOC and non-VOC lineages, VOC spike amino acid mutations are actually less likely to be included in the descendant genome. This is the opposite of the prediction for a higher rate of false positives associated with recurrent mutations. We also speculate that an excess of convergent nucleotide substitutions might actually decrease statistical power to detect true positives rather than inflate the false negative rates. The reason is that we expect that there will be more permissive placements for a sequence containing such recurrent mutations on the phylogeny which might decrease the initial placement parsimony score or decrease the partial placement parsimony score when searching the genome for alternative placements.

The concern about variation in true positive rates is also addressed in detail above. Here we emphasize that there appears to be little bias across the genome with acceptor/donor pairs selected at random and when we subset the acceptor/donor nodes to be most genetically similar. Because there appears to be little bias in the true positive rates, we conclude that a biased landscape of recombination is likely to reflect some biological reality in the detected set of recombinant nodes.

There is a wide range in false discovery rates across events associated with a specific starting parsimony score and improvement (Table 4). This makes me wonder if this should not be taken into account in the recombination identification. For example, for recombination events with a starting parsimony score of 6 and an improvement of 6, the FDR is > 0.5 . Can we ignore this and only consider the average FDR of 0.13?

This challenge reflects in large part the modest number of recombination events that are expected to be found with a parsimony score of 6 that improves by 6, and many other classes of improvement. Hence, the expected FDR might be quite high. We have addressed this concern by demonstrating that the key findings in our manuscript are supported when we only consider cases of parsimony score improvement with $FDR < 0.2$ (Text S9).

As the Authors point out, recombination detection in genetically similar viral lineages is challenging and will lead to underestimates of the overall frequency of recombination. It therefore seems remarkable that about 40% of the events are detected within Pango lineages. It is possible that this reflects for example more recombination opportunity due higher probability of co-circulation. Some lineages like B.1.1.7 may also be densely sampled offering a lot of opportunity to pick up recombinants. I think it might be useful to plot the frequency distribution of recombination with respect to the genetic distance between the donor and acceptor lineages.

We agree that this is useful and now include this plot as Figure S5, in the section Text S11: Genetic distance among donors and acceptors.

For the practitioner, it would be useful to be able to check whether data sets include any genomes that are descendants of the recombination events detected here (and filter them). This would require making available a list of the 43,163 descendant samples (and perhaps updating this if the Authors are doing real-time detection).

We agree that this is useful and now include a full list of genomes with putatively recombinant ancestry as Data S2. As described earlier in this response, we are also preparing to update the set of recombinants weekly in the future as we run RIPPLES on updated phylogenies.

Minor comments

"Additionally, excess similarity of geographic location and date metadata among the descendants of donor and acceptor nodes strongly supports the notion that the ancestors of recombinant genomes co-circulated within human populations (Supplementary Text 10, Supplementary Fig. 4) – a prerequisite for recombination."

This seems to be an overstatement -- there is a significant difference in overlap for the identified recombinants, but no strong evidence that there was co-circulation for all recombinants. There is still a considerable fraction for which there is no detectable spatial overlap for example (Fig. S4A)

We agree and we have revised the text to reflect the reviewer's comment. The text now reads:

"Additionally, excess similarity of geographic location and date metadata among the descendants of donor and acceptor nodes supports the notion that many ancestors of recombinant genomes co-circulated within human populations (Text S10-S11, Fig. S4-S5)."

Indeed, it is not possible in many cases to know whether two lineages co-circulated because we only sample the tips and not the acceptor/donor ancestors that actually gave rise to an observed recombination. We also present some explanation for this in Text S10.

"Furthermore, a recombination event that transferred a portion of the Spike protein coding region into the ancestor of SARS-CoV-2 may have contributed to the emergence of the COVID-19 pandemic in human populations 11."

This has been shown to be a misinterpretation due to the fact that the bat RaTG13 virus is significantly more divergent in the receptor binding and thus the plausible recombinant (<https://doi.org/10.1038/s41564-020-0771-4>), as also independently confirmed by (<https://doi.org/10.1093/ve/veaa098>).

We agree and we have revised this section to remove the statement that SARS-CoV-2 has recent recombinant ancestry.

SI: "As would be expected, more modest parsimony score improvements are associated with a higher estimated false discovery rate (Supplementary Table 3)." Should this not refer to Table 4?

This should. We have corrected the typo and thank the reviewer for pointing this out.

Referee #3 (Remarks to the Author):

There have been a number of recent papers and preprints (cited by the authors) investigating recombination within Sars COV-2, which are broadly consistent with each other.

This analysis presents a new method and the biggest survey of recombination to date based on looking for signals within a phylogeny generated using 1.6 million genomes. The signal used by the method to find putative recombination is long branch length in the reconstructed phylogenetic trees. For these lineages, the method makes exhaustive search of putative hybridisation events. The method works well in highly sampled organism but would likely be less reliably in poorly sampled organisms (which are the great majority) because recombination only reliably increases branch lengths in the presence of intermediate sequences. If these are absent other signals may be more effective. Since the paper claims to present a general new methodology, this issue of when it works well does need to be addressed.

In the concluding statements of the main text (page 7), we have added statements to explicitly describe where this method is expected to perform best. We also note that although many organisms are poorly sampled, we expect that the future will include many extremely densely sampled datasets such as the one that we evaluate in this work. Our method is therefore poised to facilitate interpretation of genomic variation during the pandemic and into the future.

Furthermore, the test of the method is mainly a test of whether it can identify manually generated recombination events, with a very optimistic headline figure of 93% detection rate is given. For many real world datasets (less so for COV-2 because of the density of sampling) the question of whether the position of the donor sequences in the phylogeny was accurately estimated would be very important but this aspect is not tested at all.

This method has been evaluated in detail in related works. Specifically, in Turakhia et al. (2021. Nat Gen). We showed that our approach is capable of accurately inferring phylogenies in SARS-CoV-2 by sequential placement of samples. More recently, Thornlow et al. (2021. BioRxiv), demonstrated that the optimization approach (matOptimize) that we used here produces the most accurate phylogeny of any of the methods that have been widely used during the pandemic. We now provide this explanation and citations to these works in our manuscript.

We also provide an analysis of the ability of RIPPLES to identify recombinants and when the tree was inferred de novo and is therefore subject to potential phylogenetic reconstruction inaccuracy (Text S6).

We also address this concern through simulations, detailed in our responses to questions raised by reviewers 1 and 2 as well as in Text S6.

This work really makes two claims for newsworthiness. The first is that the method is easy enough to use that it can become a routine part of analysis of viral sequences. I think this is valid in terms of useability, but with the proviso that it has not been proven to be applicable to less densely sampled organisms as described above.

We have added a description of when our method is expected to perform well—densely sampled genomic datasets and also state in the conclusion paragraph that the method has yet to be validated on other species (page 7).

The second is to be a comprehensive analysis of recombination within Sars COV2.

For example, they find 600 putative recombination events, while a recent paper of Varabyue published in Genetics finds 200. They make the specific claim that around 2.7% of SARS-Cov2 sequences are recombinant and estimate that only a very small fraction of genetic changes are currently made by recombination. I basically buy the validity of these analyses and the statistics generated are novel and are likely to be well cited.

Nothing is said about the distribution of recombination events on the tree, e.g. in terms of age distribution and frequency in particular lineages. Geographical information would also be very interesting.

We do provide an analysis of the clade sizes of recombinant lineages, which is related to the ages of each group.

We now provide an evaluation of the genetic distances among inferred donor and acceptor nodes (Text S11, Fig. S5). Briefly, our results suggest that acceptor/donor pairs are typically more closely genetically related than randomly distributed pairs of nodes. This result lends additional credibility to the idea that our approach is correctly identifying recombinant lineages.

We are hesitant to delve too deeply into the geographic origins of the putative recombinant lineages. The reason for this is that sequencing efforts vary dramatically across countries and virus lineages move quickly throughout the world. It is challenging to accurately and fairly represent these data.

The manuscript also finds that recombination is higher in the last part of the gene, where the spike protein is and finds that unsurprisingly but reassuringly recombination occurs preferentially between lineages that are found to co-occur in time and space.

The paper also claims that this is due to mechanistic factors rather than natural selection. While mechanistic factors are perfectly plausible, I don't find the arguments for this especially convincing. The selection responsible may very well be transitory in nature, e.g. due to immune selection in individual hosts and the fact that the lineages are not obviously successful is poor evidence for the absence of selection.

This is reasonable and certainly correct. We have heavily revised this paragraph (page 4) to say more narrowly that positive selection at the level of inter-host transmission probably does not play a major role in driving viral recombination.

The manuscript claims in the abstract that identifying recombination events is important from a practical point of view essentially to identify variants of concern, but I think this is very much unproven at least in the context of evolution within SARS-CoV2. Probably the lineages most likely to be VOC are those where multiple substitutions have arisen de novo, driven by selection (and epistasis). There is no evidence that any of the existing VOCs arose by recombination.

At present, there is little to suggest that major VOCs arose as a result of recombination. However, viral recombination is known to be an important driver of evolutionary dynamics in diverse lineages. While it has not been a major driver in SARS-CoV-2 yet, the ability to accurately and rapidly detect such events is crucial to interpreting and using viral genome sequences in public health. For example, using RIPPLES, it is possible to test between the possibility of a recombination event and an accumulation of mutations on a branch, which have very different evolutionary interpretations and implications. Recombination is instantaneous, while the accumulation of mutations requires time, and so the latter supports the presence of unsampled reservoirs (e.g., long-term hosts, unsampled populations of humans or animals) as the source of VOCs, while the former does not. So, while VOC's have not arisen as a result of recombination yet, the ability to accurately and rapidly assess such possibilities will be essential in the future as new VOCs emerge.

Moreover, it is fundamentally important to establish expectations for the baseline rate of recombination across the SARS-CoV-2 genome. Such expectations might reveal when new, more frequently recombinant lineages evolve. This approach therefore has a range of possibly important outcomes.

Referee #4 (Remarks to the Author):

This is an interesting paper on the role of recombination in the on going pandemic. The authors perform a very exhaustive recombination analysis, involving more than one million sequences available worldwide. For this purpose, they developed a novel method (RIPPLES) which leverages the information from an input (large) phylogenetic tree to quickly detect recombination events.

In my opinion, the novelty of this work is based on the detection of hundreds of recombination events in SARS-CoV-2, rather than on the results or approach shown here. While previous works have reported recombination previously (e.g. Jackson et al, 2021), these studies were focused on much smaller datasets and/or specific viral lineages.

We thank the reviewer for this comment. We have revised the manuscript title by taking this feedback into consideration.

However, some of the results highlighted in the manuscript are either already known (e.g. unequal distribution of recombination breakpoints in Coronaviruses have been reported in several publications) or have an unclear impact in terms of clinical or biological relevance (e.g. B.1.355 lineage being a recombinant).

Although the non-uniform rate of recombination in other coronaviruses has been suggested previously, one key distinction between our analysis and those in other coronaviruses is the time-scale considered and the relevant species under consideration. It is essential that we determine whether biologically similar phenomena are occurring in SARS-CoV-2 and other coronaviruses. Additionally, in SARS-CoV-2 each recombination event must be extremely recent and for reasons we have now expanded upon in the text (see also our response to reviewer 3), there is little reason to expect that selection enhancing inter-host transmission is the primary driver of this difference. This type of insight necessarily requires the analysis of many closely related genomes.

Additionally, the ability to accurately identify recombinant lineages in nearly real time is important both for establishing the expected frequencies of recombination (and whether this evolves through time) and to accurately determine if a future VOC is likely to be a recombinant.

In addition, I have the following specific major comments:

- It is noteworthy that one of the problems when detecting recombination is the lack of power to detect it at low levels of genetic diversity. Indeed, recombination between closely related sequences does not necessarily increase diversity/branch lengths. This method does not address this limitation. It only focuses on recombination events that cause the generation of branches of longer length. Thus, the detection of recombination events may be skewed towards those events occurring between relatively distant parental lineages.

This is correct. We have highlighted this limitation more prominently in the main text and expanded our investigation in the supplemental materials to accurately delineate our detection approach. However, we emphasize that recombination among the most genetically dissimilar acceptor/donor pairs are also the most likely to generate new viral lineages with novel phenotypes. These are precisely the recombinant lineages that should be the primary focus of downstream analyses.

- Sup Figures 1 and 2 show some plots on the performance of RIPPLES. I cannot find the study showing a rate of false positives, and negatives, under different scenarios of i) genetic variability, ii) recombination rate (expected number of recombination breakpoints found in the simulation), iii) sample size of the dataset. Very importantly, this work lacks a comparison with pre-existing methods for the detection of recombination (RDP, 3seq, Bootscan, etc).

The considerations raised in (i, ii) are now addressed more deeply in our supplemental materials as well as the responses to reviewers 1, 2, 3 (above).

We now also provide a detailed comparison to existing methods focusing in particular on the computational requirements, runtimes and p-values that can reasonably be obtained in Text S18. As it is infeasible to run any existing approach on the full dataset, we extrapolate what might be required to do so.

We have compared RIPPLES to RDP and 3seq, among other methods, and discuss these results in Text S18. In summary, RIPPLES takes less than one second to exhaustively search for evidence of recombination for one sample in a 1,000-sample phylogeny, while RDP takes more than one hour

for the same comparison. We therefore conclude that RDP was prohibitively slow for any meaningful accuracy comparison on our full dataset.

We have also compared RIPPLES to 3seq using simulated recombinants. We selected two large clades at random from our dataset, and then selected 1,000 random samples from each, and created a tree from scratch using just these 2,000 samples using UShER. We then created 500 recombinants by selecting one random sample from each clade and a random breakpoint. To test RIPPLES, we separately placed each simulated recombinant on the 2,000-sample tree and used RIPPLES to search for evidence of recombination on the now 2,001-sample tree. RIPPLES finds the proper breakpoint for 410 of the 500 simulated recombinants when tested in this way.

When using 3seq to examine the 500 trios individually and specifying the two parents and simulated recombinant child, 3seq finds the proper breakpoint for 491 of the 500 trios. However, testing in this way would be impossible unless the identity of the recombinant sample and its two parents were already known in advance. When we use 3seq to examine the entire 1289-sequence dataset (500 simulated recombinants and 1,000 parents each, chosen with replacement), 2.13×10^9 triplets are examined and a p-value of 2.34×10^{-11} is required for rejection of the null hypothesis, and 0 recombinants are detected. The nature of exhaustive recombination searches via trios fundamentally prevents detection of recombination without prior knowledge of the recombinant sample and its parents due to the need for extensive p-value corrections or extensive subsampling. However, we emphasize that *post hoc* application of 3seq and other programs as we do in our work is a valuable partially orthogonal way to evaluate evidence for recombination in nodes detected by RIPPLES.

- Finally, the authors claim that this method is very fast. However, I understand that this was calculated from a starting point that includes a precomputed phylogenetic tree as input. Thus, we need to add the time of tree inference to computational time? Also, these computational times have been measured running RIPPLES in really large computers (224 vCPUs). Would it be possible to analyze these SARS-CoV-2 samples in a smaller computer?

Currently, it is feasible to infer and optimize a 1.6M sample tree in approximately 3 days. The combined time of inference, optimization, and search for recombinants would therefore be substantially less than trio-based approaches. However, we do not consider tree-inference to be a part of the run time requirements. Online phylogenomics, of which RIPPLES is one example, is a framework wherein large, heavily optimized phylogenies are readily available for heavily sampled organisms such as SARS-CoV-2 (Thornlow et al., bioRxiv 2021). In the intended usage of RIPPLES, users will be able to download a nearly comprehensive phylogenetic tree, add their samples, and determine whether they might contain recombinant ancestry very quickly. This expected usage is described in detail in the supplemental text and we now add a description of the expected runtime requirements.

It is certainly possible to run RIPPLES as described on a much smaller computer. For example, using a basic laptop and in approximately 5 minutes, one can download the public phylogeny inferred using USHER, add 50 samples (using USHER), and search specifically those lineages for recombinant ancestry in RIPPLES. RIPPLES is designed in such a way that one can easily focus recombination efforts on the portion of the tree that a user has modified.

Minor comments:

- Authors should specify clearly how they define genetic distance. Number of nucleotide differences? Did they use any specific substitution model?

We use the number of nucleotide differences. We have added this definition to the manuscript.

- Regarding the performance: Is the false discovery rate of 11.6% affected by any of the post hoc filters mentioned in the manuscript?

We are computing that FDR after applying the post hoc filters, as such the rate is not affected. Our key results from the manuscript are also not affected by application of an FDR-based filter. We now clarify this in Text S9.

Reviewer Reports on the First Revision:

Referees' comments:

Referee #1 (Remarks to the Author):

I am happy with the authors' response to the reviewers' comments regarding possible ascertainment bias related to their finding of a higher rate of recombination in the 3' end of the genome. I think that my specific concern regarding a potential higher rate of true positives in regions of the genome with higher genetic diversity is addressed by Supplementary Figure S1 and associated methods. I slightly wish that the authors had exactly replicated their main Figure 2A using the simulated data for ease of comparison, but I am not insisting on this.

I asked if the authors had considered whether inferring recombination based on a phylogeny which itself has been affected by recombination might be problematic. I probably muddied the waters by making a second, more vaguely worded, request in the suggested improvements section to consider how the method would be affected by the "correctness" of the input tree. I meant the same thing as in my original comment here- to what extent might representing a network using a single evolutionary history affect their method? I understand that the method identifies putative recombinants for testing based on the artefactually long branch lengths that result from treating the network as a single tree. What I wanted to know was whether/how the network-as-tree representation might affect the search for donors and acceptors. I'm not sure I have good intuition for this. As the method requires a monolithic phylogeny including the putative parents to test against, and this phylogeny will become increasingly less strictly correct as more recombination happens between more diverse clades in the future, and the authors are positioning this software as a tool to detect recombination in SARS-CoV-2 and other things as we move forward, this is relevant. The authors have not satisfactorily addressed this point.

The pandemic is 10 months further along than the dataset that analysed here, with something like 10 million sequences now available on GISAID. The authors might qualify the 1.6 million figure in the introduction with the date that this sample pertains to, because it will affect the interpretation of the 2.7% figure of samples with recombinant ancestry presented in the same paragraph. Relatedly, they might update the second paragraph of the Main Text (lines 55-65), given the several omicron-delta and omicron-omicron recombinants that have been described recently?

I think the ~two-order-of-magnitude speedup of the program detailed in the rebuttal is great. Do lines 200-201 need updating – with the speedup and/or with reference to the 1.6 million sample vs. the current phylogeny?.

Some more minor comments below.

Lines 230-231 + Text S6 could be a bit clearer regarding the breakdown of the 8,000 simulated recombinants ($\{1, 2\}$ breakpoints * $\{0,1,2,3\}$ extra mutations * $\{1000\}$ pairs of parents = 8000 ?)

Supplementary Text Lines 170-171– what proportion of all the simulated recombinants are present in the set of which 84% are recovered?

Supplementary Text Line 352– what is the direction of the significant correlation?

Figure S1 – the caption should better explain this figure – in particular, what do the three colours mean (excess, deficit, same proportion).

Jackson et al. 2021 is published, but the authors cite the medrxiv version.

Referee #2 (Remarks to the Author):

The Authors have adequately addressed my concerns. I have no further comments.

Referee #3 (Remarks to the Author):

I thank the authors for their thorough and sensible responses to my and other reviewer comments. I have no additional concerns.

Referee #4 (Remarks to the Author):

The authors have addressed the comments.

Author Rebuttals to First Revision:

Referees' comments:

Referee #1 (Remarks to the Author):

I am happy with the authors' response to the reviewers' comments regarding possible ascertainment bias related to their finding of a higher rate of recombination in the 3' end of the genome. I think that my specific concern regarding a potential higher rate of true positives in regions of the genome with higher genetic diversity is addressed by Supplementary Figure S1 and associated methods. I slightly wish that the authors had exactly replicated their main Figure 2A using the simulated data for ease of comparison, but I am not insisting on this.

We appreciate the reviewer's careful consideration of our response and efforts to improve the robustness of the conclusions that we draw in our work.

I asked if the authors had considered whether inferring recombination based on a phylogeny which itself has been affected by recombination might be problematic. I probably muddied the waters by making a second, more vaguely worded, request in the suggested improvements section to consider how the method would be affected by the "correctness" of the input tree. I meant the same thing as in my original comment here- to what extent might representing a network using a single evolutionary history affect their method? I understand that the method identifies putative recombinants for testing based on the artefactually long branch lengths that result from treating the network as a single tree. What I wanted to know was whether/how the network-as-tree representation might affect the search for donors and acceptors. I'm not sure I have good intuition for this. As the method requires a monolithic phylogeny including the putative parents to test against, and this phylogeny will become

increasingly less strictly correct as more recombination happens between more diverse clades in the future, and the authors are positioning this software as a tool to detect recombination in SARS-CoV-2 and other things as we move forward, this is relevant. The authors have not satisfactorily addressed this point.

The reviewer raises an important question regarding the degree to which recombination affects the inferred phylogeny. Because recombination in SARS-CoV-2 is relatively rare and because most recombinants inherit the majority of the genome from one parent, it is possible to infer a "clonal frame" as is commonly done in bacteria. Doing this can result in an inaccuracy in that many mutations are represented as a long branch (indeed, this is the signal we exploit with RIPPLES to find potential recombinants), but phylogenetic inference in itself is often relatively robust (Hedge

and Wilson 2014). Even when recombination events are represented as long branches, the parents will still be represented correctly in the clonal frame. We now describe this idea, and include the paper referenced above, in the main text where we explain the core of our method.

We emphasize that for all simulations including *de novo* inference, recombinants are typically found individually on long branches (85% of cases) consistent with the explanation that their inclusion creates a single additional long branch without substantially affecting inference for other non-recombinant samples because those samples are not drawn onto the long branch that's artifactually produced by accommodating recombination within a single phylogeny.

To further evaluate whether our approach of representing a network as a single tree introduces an unacceptable excess of phylogenetic inference errors, we analyzed simulation experiments as we described in our resubmission. Briefly, these experiments included generating a single recombinant and inferring the tree *de novo* for a subset of 2,000 samples. These simulations are conservative, because the rate of recombination events per sample ($1/2000 = 0.0005$) is slightly larger than we estimate for the empirical analysis of the full dataset ($589/\sim 1.6e6 = 0.00037$). If including recombinants during phylogenetic inference induces an excess of errors in the resulting topology, we would expect that the topology would differ for the non-recombinant samples between inferred trees that include or exclude the recombinant lineage. To determine the expected difference between two independently inferred trees without the impact of recombination, we randomized the input order of samples and inferred a second phylogeny without including the recombinant lineage. The idea is that this is a null model for how different we would expect two trees to be purely due to stochastic differences in phylogenetic inference. We then evaluated differences between trees using the Robinson-Foulds distance to compare pairs of trees. That is, in our analysis, there are two sets of RF distances:

1. Between two trees with identical input samples but distinct input orders.
2. Between trees produced including one recombinant and trees inferred without including the recombinant. We pruned the simulated recombinant before comparing the inferred topologies to ensure the sample sets were identical.

We found that the RF distances between topologies inferred including the recombinant lineage (set 2) are not significantly greater than those inferred without including the recombinant (set 1) during phylogenetic inference ($P > 0.38$, Wilcoxon Test). This implies that the effect of including a recombination lineage on *de novo* phylogenetic inference is minimal and the tree topology is not dramatically affected by the inclusion of recombinant samples. Note that although these are smaller trees than the larger phylogeny we analyzed here, because we include the most closely related samples to each recombinant's ancestors in phylogenetic inference, we expect that the impacts are captured well for trees of equivalent sampling density—even those that are substantially larger.

We therefore conclude that so long as recombination is relatively uncommon, the impact on the inferred phylogeny will be minimal. Our analysis suggests that this is indeed the case and that we can reliably detect recombinants using our approach.

The pandemic is 10 months further along than the dataset that analysed here, with something like 10 million sequences now available on GISAID. The authors might qualify the 1.6 million figure in the introduction with the date that this sample pertains to, because it will affect the interpretation of the 2.7% figure of samples with recombinant ancestry presented in the same paragraph. Relatedly, they might update the second paragraph of the Main Text (lines 55-65), given the several omicron-delta and omicron-omicron recombinants that have been described recently?

This is a good point. We now state in the introductory paragraph that the phylogeny used for this analysis includes all whole genome sequence data available through May of 2021. This context is important for understanding our results.

We believe the second paragraph of the main text is correct as written. In that most inter-lineage recombinants (outside of this manuscript) have been discovered via ad hoc or methods that reduce the search space to consider recombinants between specific predefined nodes in the phylogeny.

I think the ~two-order-of-magnitude speedup of the program detailed in the rebuttal is great. Do lines 200-201 need updating – with the speedup and/or with reference to the 1.6 million sample vs. the current phylogeny?.

They do, and the runtimes have now been updated in the document.

Some more minor comments below.

Lines 230-231 + Text S6 could be a bit clearer regarding the breakdown of the 8,000 simulated recombinants ($\{1, 2\}$ breakpoints * $\{0,1,2,3\}$ extra mutations * $\{1000\}$ pairs of parents = 8000 ?)

This text in the methods and Supplemental text S6 has been updated to improve clarity and to resolve typos. In our revised simulations (in response to the first round of reviews) we created 2,000 simulations, and have now revised Text S6 to reflect this.

Supplementary Text Lines 170-171– what proportion of all the simulated recombinants are present in the set of which 84% are recovered?

RIPPLES detects 75.8% of all recombinants, without conditioning on their initial parsimony score. This has now been clarified in the supplemental text.

Supplementary Text Line 352– what is the direction of the significant correlation?

This is a positive correlation and we now clarify that in the supplemental text.

Figure S1 – the caption should better explain this figure – in particular, what do the three colours mean (excess, deficit, same proportion).

The caption has now been updated to provide an explanation of the differences.

Jackson et al. 2021 is published, but the authors cite the medrxiv version.

This has been corrected.

Referee #2 (Remarks to the Author):

The Authors have adequately addressed my concerns. I have no further comments.

Referee #3 (Remarks to the Author):

I thank the authors for their thorough and sensible responses to my and other reviewer comments. I have no additional concerns.

Referee #4 (Remarks to the Author):

The authors have addressed the comments.

We thank all reviewers for their careful attention to our work and helpful comments.

Reviewer Reports on the Second Revision:

Referees' comments:

Referee #1 (Remarks to the Author):

I am happy with the authors' responses to my comments.